# Simulation modeling to assess performance of integrated healthcare systems: Literature review to characterize the field and visual aid to guide model selection

Nicolas Larrain [1,2]*, Oliver Groene[1]

**1** OptiMedis AG, Hamburg, Germany, **2** Hamburg Centre for Health Economics, University of Hamburg, Hamburg, Germany

* n.larrain@optimedis.de

## Abstract

### Background

The guiding principle of many health care reforms is to overcome fragmentation of service delivery and work towards integrated healthcare systems. Even though the value of integration is well recognized, capturing its drivers and its impact as part of health system performance assessment is challenging. The main reason is that current assessment tools only insufficiently capture the complexity of integrated systems, resulting in poor impact estimations of the actions taken towards the 'Triple Aim'. We describe the unique nature of simulation modeling to consider key health reform aspects: system complexity, optimization of actions, and long-term assessments.

### Research question

How can the use and uptake of simulation models be characterized in the field of performance assessment of integrated healthcare systems?

### Methods

A systematic search was conducted between 2000 and 2018, in 5 academic databases (ACM D. Library, CINAHL, IEEE Xplore, PubMed, Web of Science) complemented with grey literature from Google Scholar. Studies using simulation models with system thinking to assess system performance in topics relevant to integrated healthcare were selected for revision.

### Results

After screening 2274 articles, 30 were selected for analysis. Five modeling techniques were characterized, across four application areas in healthcare. Complexity was defined in nine aspects, embedded distinctively in each modeling technique. 'What if?' & 'How to?' scenarios were identified as methods for system optimization. The mean time frame for performance assessments was 18 years.

**Funding:** This project has received funding from the European Union's Horizon 2020 research and innovation programme under the Marie Skłodowska-Curie grant agreement No 765141. Both authors are employed at OptiMedis AG. The funder provided support in the form of salaries for authors NL & OG, but did not have any additional role in the study design, data collection and analysis, decision to publish, or preparation of the manuscript. The specific roles of these authors are articulated in the 'author contributions' section.

**Competing interests:** The authors declare no conflict of interest. Both authors are employed at OptiMedis AG, a commercial company. This does not alter our adherence to PLOS ONE policies on sharing data and materials.

## Conclusions

Simulation models can evaluate system performance emphasizing the complex relations between components, understanding the system's adaptability to change in short or long-term assessments. These advantages position them as a useful tool for complementing performance assessment of integrated healthcare systems in their pursuit of the 'Triple Aim'. Besides literacy in modeling techniques, accurate model selection is facilitated after identification and prioritization of the complexities that rule system performance. For this purpose, a tool for selecting the most appropriate simulation modeling techniques was developed.

## 1. Introduction

The guiding principle of many health care reforms is to overcome fragmentation of service delivery and work towards integrated healthcare systems (IHS) [1, 2]. Integrated healthcare comes in the form of linkage and coordination of providers along the continuum of care [3]. By focusing on the nature and strength of the links between the system components, IHS rely on and enhance the complexity of the health system [4] to achieve a threefold objective; improve health of the population, improve patient (and carer) experience while reducing healthcare costs ('Triple Aim') [5]. To reach the Triple Aim, IHS introduce solutions to ensure the sustainability of health care provision through investment in preventive care and constant improvements in clinical practice [3].

IHS success has been evidenced in numerous publications. However, healthcare managers encounter problems when assessing the drivers of this success [6–8]. These challenges arise because assessment tools are not specific to integrated care and don't consider the unique nature of the approach [9, 10]. Acknowledging a lack of specific assessment tools, the Expert Group on Health System Performance Assessment of the EU created a standard framework for performance assessment of integrated care [3, 6, 7]. The framework consists of a series of key performance indicators in topics relevant to IHS. However, even though specific to IHS, the indicators compiled by the expert group are insufficient to capture the full value of integrated care. The problem is not in the completeness of the indicator list, but in monitoring indicators as a performance assessment approach. Indicators are developed based on assumptions about the interrelation between a measure and the system objectives [11], but the causal pathways are not described and are known to be multiple, non-linear, with changing causal effects and affected by several individuals and contextual factors [12, 13]. In other words, the traditional approach can't capture the complexity of the health system [4]. Because IHS enhance system complexity and strive for efficiency and accountability in every component of the system, indicators alone are insufficient to guide improvement [9, 12].

Complex health policy issues can be better assessed with methods that enable research synthesis and utilize a complex systems perspective [14, 15]. As defined by Petticrew et al. (2018) [15]; a complex system perspective (or just 'systems thinking') is defined by acknowledging the value arising from the relationships between the system components and their dynamic properties. When used for evaluating system performance, system thinking answers an essential concern for IHS 'how did the intervention reshape the system in favorable ways?'.

Simulation modeling (SM) is a discipline with the features necessary to implement systems thinking when assessing performance of a health system [14, 16, 17]. A simulation is a virtual recreation of a real system. It is used to test situations and understand the effect of

interventions on the performance of the system over time. Combining expert opinion with observational and experimental results, SM provides a relatively inexpensive way to estimate individual and population-level effects of changes in the system's determinants of performance.

There is extensive literature reviewing simulation models in the healthcare sector. Salleh et al. [18] published an umbrella review including 37 reviews, that together cover articles from 1950 to 2016 and explore the wide range of applications in healthcare, software tools and data sources used in the field of healthcare simulation. Meanwhile, the paper by Günal & Pidd [19] starts by narrating the historic progression of simulation modeling an its applications in healthcare, giving some idea of the long history of the field. Most recently (2021), Roy et al. [17] analyzed healthcare simulation literature of the past decade, addressing issues in various healthcare service delivery levels and categorizing the literature accordingly. Altogether, literature in the field provides a comprehensive characterization of simulation models in healthcare, including; the areas and types of application where the discipline has been used, the techniques available, data sources, simulation software [18, 20, 21], type of outputs and level of insight, inputs and resources required [22], relative frequency of use and level of implementation [23] and specific aspects of a care facility operations where techniques are most common [17, 24, 25]. These topics are most commonly analyzed following a structure similar to the one best represented by Mielczarek et al. [25], who creates a system of classification of health care topic areas assessed with simulation methods. The objective is to investigate the usefulness of modeling techniques and their correlation with a corresponding health care application. While authors add innovations to this common structure, such as the identification of research gaps influencing the limited uptake of the discipline [20, 23, 24, 26] or exploring the link between interventions and key performance indicators (KPI) [26], complexities in the relationships of system components have been heavily underassessed. Roy [17] recognizes the complexity of the health system and the ability of simulation modeling to address this complexity, but his review focuses on capturing specific health issues addressed, operations management concepts applied, simulation methods used, and identifying major research gaps—a framework similar to the one by Mielczarek et al. [25]. Vanbrabant et al. [26] also acknowledges simulations as the technique most suitable to capture the randomness and complexity of patient flow through the emergency department. But the analysis is limited to providing insights into which interventions influence which KPI. In the same line, Laker [27] also recognizes the usefulness of simulation models to integrate complexity, and provides an excellent summary of the properties of four simulation techniques. However, it fails on providing a common framework to characterize and contrast the complexities that can be represented in each technique. Complexity is also indirectly mentioned in identified research gaps, when both Vanbrand et al. and Yousefi et al. [20] state the underuse of simulation models in multi-objective evaluations and Brailsford et al. and Roy et al. [17, 24] suggests that healthcare is an area of application for hybrid simulation due in part to increasing system complexity.

By overlooking complexity, the advantages of simulation modeling and the challenges of IHS performance assessment remain unmatched. Furthermore, simulation time frames and optimization capabilities, standard knowledge for simulation experts but not for healthcare managers [28], are also overlooked in reviews summarizing the use of SM in healthcare. The gap results in simulation models not been systematically picked up by integrated healthcare managers to assess performance of IHS. The issue was partially addressed in 2015 by the ISO-POR task force [14, 29], who published a series of papers describing how three of the most common simulation modeling techniques can be used to evaluate complex health systems and provide descriptions and tools to implement them accurately. However, at the time there was no common understanding of the drivers affecting IHS performance hence a clear explanation

and exemplification of how these particularly complex health systems could make use of simulation models to assess performance was not possible.

This literature review is intended to bring together the field of performance assessment of integrated healthcare systems and the discipline of simulation modeling. We contribute to the vast literature characterizing the use of simulation modeling in health system performance assessment by focusing specifically on the discipline's ability to implement a complex system perspective in topics relevant to IHS. Our research is directed to readers that seek to expand performance assessment tools while considering the enhanced complexity embedded in the integrated care approach. We conclude our analysis with the creation of a practical tool for selecting the most appropriate simulation modeling technique depending on the characteristics of the system to be modeled.

## 2. Methods

### 2.1 Search strategy

A comprehensive search strategy was performed directed to find articles that allowed us to understand how simulation modeling techniques implemented a complex system perspective in topics relevant to IHS. The systematic search was conducted in 5 academic databases (ACM Digital Library, CINAHL, IEEE Xplore, PubMed & Web of Science). Grey literature was searched for in Google Scholar and only considered if articles complied with all the criteria in the AACODS checklist for critically appraising grey literature [30]. Finally, papers were also added through snowballing. The search was conducted for the period 01/01/2000–31/12/2018 as an increased interest in SM has been documented after this starting date, supported by technology advances [18]. The review was registered in PROSPERO (Registration number: CRD42020149658).

A Boolean search code was developed with three scopes of terms. The first scope, "*Technique*", filters for simulation modeling techniques and combines 17 systematic search strategies extracted from the umbrella review by Salleh et al. [18] added to the list of simulation modeling techniques described in Jun et al. [22]. The second scope, "*Integrated healthcare systems topics of interest*" is defined by 76 search terms, extracted from the indicator types and domains stated in the framework for performance assessment of IHS developed by the Expert Group of the European Commission [3, 6, 7], the systematic review of methods for IHS performance assessment by Strandberg-Larsen et al. [31] and the "Care Coordination Measures Atlas" by McDonald et al. [32]. Finally, the third scope refers to the healthcare sector. Terms in the first scope ("*Techniques*") were restricted to the title, and terms in the other scopes were restricted to title/abstract. The complete list of terms can be found in S1 Table.

### 2.2 Selection criteria

**Inclusion criteria.** Only health system evaluations taking a complex system perspective were considered. Furthermore, we only included articles that used a simulation model in the list of techniques described in Salleh et al. [18] or in Jun et al. [22] as SM techniques or self-identified as such. Finally, articles further had to address the performance assessment of an IHS topic-of-interest. The lists of SM techniques and IHS topics of interest can be found in S2 Table.

**Exclusion criteria.** We excluded studies that described non-computer-based simulation models. Also, we excluded studies that were not calibrated and validated against data from a real situation. Finally, we excluded from the data extraction and analysis studies whose reporting standards were insufficient to replicate the assessment or did not fully enable reviewers the complete understanding of the implementation of systems thinking. To comply with the latter criteria, only studies graded 'A' in quality assessment were selected.

### 2.3 Quality assessment

Two independent reviewers (SW & NL) assessed the quality of papers during the screening process. Using the quality assessment tool developed by Fone et al. [33] to appraise simulation modeling studies, reviewers gave a score of 0, 1, or 2 in ten criteria and created four quality groups (A to D). The quality assessment was followed with an assessment of the credibility and relevance of the articles for the purpose of this review and aided reviewers to select articles for revision. Given the focus of the review, an assessment of the risk of bias in the study's results was not considered.

### 2.4 Data extraction and analysis

Data extraction was made by the main author (NL), based on the template used by Brailsford et al. in their analysis of simulation and modeling techniques for healthcare [23]. The final extraction sheet was modified focusing on two main topics. First, to characterize the different modeling techniques, their area of application, key features for implementation, together with data requirements and outcomes. Second, to characterize the complex aspects of the health system that each technique can represent. The detailed data extraction sheet can be found in S3 Table.

The analysis was conducted in two phases. First, using a 'Deductive a priori template approach' [34] articles were classified and characterized according to previous assessments of SM made by Jun et al. [22], Salleh et al. [18], and Rueckel et al. [21]. Subsequently, in a 'Data-driven inductive approach' [35], simulation modeling techniques were re-characterized in five items following the objectives of this review. Item (1.) presents the IHS topics-of-interest where SM has been successfully applied. The item aims to inform and exemplify in what situation of interest to IHS can the discipline be useful, in a similar structure of the analysis of previous literature. Even though the selection of articles is primarily intended to understand simulation models' ability to integrate the shortcomings of IHS performance assessment, and not to identify the link between simulation technique and helathcare area, a similar analysis to that of previous literature will allow us to validate our findinds when compared to conclusions of other authors. In item (2.) we supported the analysis of the reviewed papers with further literature and present an introductory description of the identified simulation techniques, explaining how they are applied in the topics of interest to IHS. The last three items were selected to explore how simulation models deal with challenges that are particularly harmful to integrated care and are not yet mirrored in traditional assessment tools [8]. Item (3.) presents the modeled complexities in relationships between system components, summarized per modeling technique. The item allows us to understand the capacities of each technique to correctly model causality paths and co-existing effects, essential concerns for several integrated care interest areas [8, 10]. Item (4.) presents the identified optimization capabilities, essential function of any tool guiding healthcare to the Triple Aim [3, 10, 11, 36]. Preventive medicine and overall population health improvements are known to show effects only after several years after intervention [7] and as they comprise an essential part of IHS, Item (5.) presents the time frame of the selected papers to understand the capacity for long-term assessments.

## 3. Results

The search resulted in 2271 unique articles. Screenings at title/abstract and full text were made by two separate reviewers (SW & NL) and resulted in seventy-six articles selected for quality assessment. Out of these, thirty studies were included for data extraction and analysis because of their reporting quality and detailed description of system thinking. Fig 1 presents the PRISMA diagram of the selection process. Selected papers are described in Table 1.

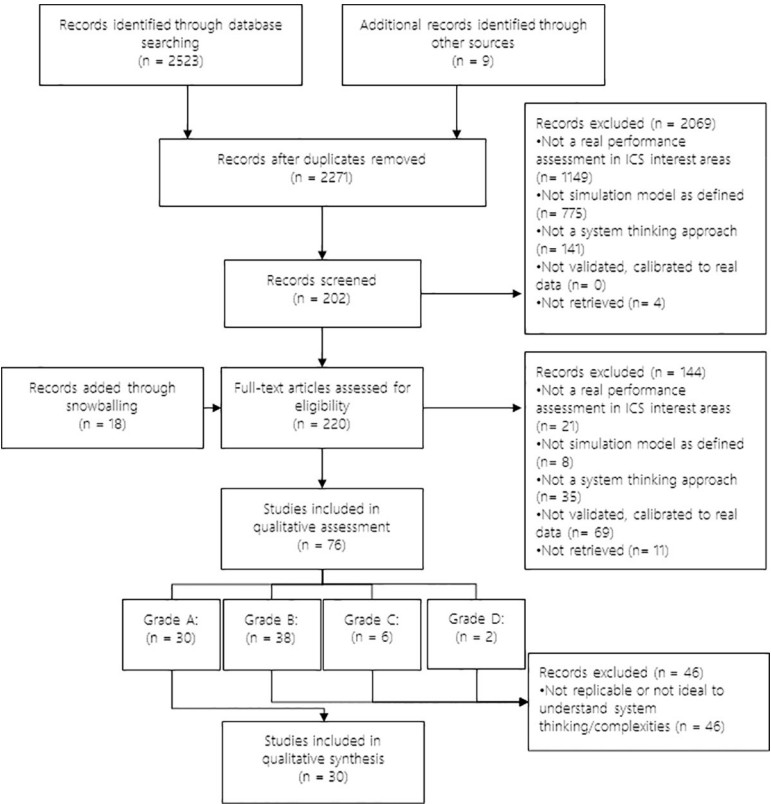

**Fig 1. PRISMA diagram.**

### 3.1 IHS topics-of-interest

Eleven IHS topics-of-interest were identified and classified in four areas of assessment. The first area of assessment covers simulation models of Policy and Strategy. This comprises studies that use simulation modeling for evaluating health policies and interventions directed to change or improve the structure, assess incentives, goals, or values in the overall system; such as (1.) pay for performance incentive scheme or (2.) national health reform evaluation. The second area of assessment covers Chronic Disease Management. Studies in this area evaluated the effectiveness of interventions or the evolution of chronic conditions, such as (3.) evaluating care management and interventions of chronic conditions and (4.) diabetes population dynamics. The third area of assessment addresses Lifestyle Interventions, including evaluation of interventions directed at lifestyle behavior, health risks, and social determinants of health, such as (5.) tobacco harm policies, market control, and interventions or (6.) evaluation of public health interventions. The last area of assessment addresses Health Resource Management and comprises studies that use SM for resource management or system design to optimize healthcare service flow or forecast demands. In this area topics were (7.) performance evaluation of community health, (8.) performance measures evaluation in outpatient center, (9.) health facility operations simulation, (10.) planning health force, and (11.) evaluation of information systems.

### 3.2 Description of simulation modeling techniques in IHS

Five simulation modeling techniques were identified in the selected articles: Two Markov Models (MM), eleven System Dynamics (SD), two Micro-Simulations (MS), twelve Discrete

**Table 1. Selected papers.**

| Author | Title | IC topic-of-interest | Model | Aspects of complexity | Optimization | Time frame |
|---|---|---|---|---|---|---|
| Alonge et al. 2017 [37] | Improving health systems performance in low- and middle-income countries: a system dynamics model of the pay-for-performance initiative in Afghanistan. | Pay for performance incentive scheme | SD | ˚ Dynamism<br>˚ Soft variables<br>˚ Interaction | 'What if'; 'How to' scenarios | 5 to 8 years |
| Ansah et al. 2016 [38] | Projecting the effects of long-term care policy on the labor market participation of primary informal family caregivers of elderly with disability: insights from a dynamic simulation model. | Evaluating care management and interventions of chronic conditions—Performance evaluation of community health | SD | ˚ Dynamism<br>˚ Influence of historical occurrence<br>˚ Interaction | 'What if' scenarios | 17 years |
| Comans et al. 2017 [39] | The development and practical application of a simulation model to inform musculoskeletal service delivery in an Australian public health service | Health facility operations simulation—Planning Health force | DES | ˚ Individualization<br>˚ Influence of historical occurrence<br>˚ Interference<br>˚ Interaction | 'What if' scenarios | 5 years |
| Cooper et al. 2008 [40] | Use of a coronary heart disease simulation model to evaluate the costs and effectiveness of drugs for the prevention of heart disease | Evaluating care management and interventions of chronic conditions. | DES | ˚ Individualization<br>˚ Influence of historical occurrence<br>˚ Interference<br>˚ Simultaneity of events<br>˚ Interaction<br>˚ Dynamism | 'What if' scenarios | 20 years |
| de Andrade et al. 2014 [41] | System Dynamics Modeling in the Evaluation of Delays of Care in ST-Segment Elevation Myocardial Infarction Patients within a Tiered Health System. | Evaluating care management and interventions of chronic conditions. | SD | ˚ Dynamism<br>˚ Interaction<br>˚ Influence of historical occurrence | 'What if' scenarios | One care case: ~4hr |
| Fialho et al. 2011 [42] | Using discrete event simulation to compare the performance of family health unit and primary health care center organizational models in Portugal. | Performance evaluation of community health | DES | ˚ Individualization<br>˚ Influence of historical occurrence<br>˚ Interference<br>˚ Interaction<br>˚ Dynamism | 'What if'; 'How to' scenarios | 1 week (1/52 year) |
| Gao et al. 2013 [43] | Tripartite hybrid model architecture for investigating health and cost impacts and intervention tradeoffs for diabetic end-stage renal disease | Evaluating care management and interventions of chronic conditions. | Hybrid | ˚ Dynamism<br>˚ Soft Variables<br>˚ Intelligent Adaptation<br>˚ Simultaneity of events<br>˚ Influence of historical occurrences<br>˚ Interaction<br>˚ Individualization | 'What if' scenarios | 1 year |
| Getsios et al. 2013 [44] | Smoking cessation treatment and outcomes patterns simulation: a new framework for evaluating the potential health and economic impact of smoking cessation interventions. | Tobacco harm policies. Market Control and Interventions | DES | ˚ Individualization<br>˚ Influence of historical occurrence<br>˚ Interaction<br>˚ Dynamism | 'What if' scenarios | Lifetime (since start smoking) |
| Goldman et al. 2004 [45] | Projecting long-term impact of modest sodium reduction in Los Angeles County | Evaluation of Public health intervention | Micro | ˚ Individualization<br>˚ Influence of historical occurrence<br>˚ Interaction | 'What if'; 'How to' scenarios | 45 years* |

*(Continued)*

**Table 1.** (Continued)

| Author | Title | IC topic-of-interest | Model | Aspects of complexity | Optimization | Time frame |
|---|---|---|---|---|---|---|
| Günal et al. 2011 [46] | DGHPSIM: Generic Simulation of Hospital Performance | Health facility operations simulation | DES | ˚ Individualization ˚ Interference ˚ Interaction ˚ Dynamism ˚ Influence of historical occurrence | 'What if'; 'How to' scenarios | 2 years |
| Hill et al. 2017 [47] | A system dynamic modeling approach to assess the impact of launching a new nicotine product on population health outcomes. | Tobacco harm policies. Market Control and Interventions | SD | ˚ Dynamism ˚ Soft variables ˚ Interaction | 'What if' scenarios | 50 years |
| Homer et al. 2010 [48] | Simulating and Evaluating Local Interventions to Improve Cardiovascular Health | Evaluating care management and interventions of chronic conditions. | SD | ˚ Dynamism ˚ Soft variables ˚ Interaction | 'What if' scenarios | 50 years |
| Jones et al. 2006 [49] | Understanding diabetes population dynamics through simulation modeling and experimentation. | Diabetes Population Dynamics | SD | ˚ Dynamism ˚ Soft variables ˚ Interaction | 'What if' scenarios | 46 years |
| Kalton et al. 2016 [50] | Multi-Agent-Based Simulation of a Complex Ecosystem of Mental Health Care. | Health facility operations simulation | ABM | ˚ Individualization ˚ Simultaneity of events ˚ Influence of historical occurrence ˚ Interaction ˚ Emergence ˚ Dynamism | 'What if'; 'How to' scenarios | 3 years |
| Kang et al. 2018 [51] | A system dynamic approach to planning and evaluating interventions for chronic disease management | Evaluating care management and interventions of chronic conditions. | SD | ˚ Dynamism ˚ Influence of historic occurrences ˚ Interaction ˚ Soft variables | 'What if'; 'How to' scenarios | 10 years |
| Kotiadis 2006 [52] | Extracting a conceptual model for a complex integrated system in health care | Health facility operations simulation | DES | ˚ Individualization ˚ Interaction ˚ Interference | 'What if'; 'How to' scenarios | 5 months |
| Laurence et al. 2016 [53] | Improving the planning of the GP workforce in Australia: a simulation model incorporating work transitions, health needs, and service usage. | Planning Health force | Markov | ˚ Interaction | 'What if'; 'How to' scenarios | 10 years |
| Lay-Yee et al. 2015 [54] | Determinants and disparities: a simulation approach to the case of child health care. | Performance evaluation of community health | Micro | ˚ Individualization ˚ Influence of historical occurrence ˚ Dynamism ˚ Interaction | 'What if' scenarios | 10 years |
| Lebcir et al. 2017 [55] | A discrete event simulation model to evaluate the use of community services in the treatment of patients with Parkinson's disease in the United Kingdom. | Performance evaluation of community health | DES | ˚ Individualization ˚ Influence of historical occurrence ˚ Interference ˚ Interaction ˚ Dynamism ˚ Simultaneity of events | 'What if'; 'How to' scenarios | 3 years |
| Levy et al. 2016 [56] | Estimating the Potential Impact of Tobacco Control Policies on Adverse Maternal and Child Health Outcomes in the United States Using the SimSmoke Tobacco Control Policy Simulation Model. | Tobacco harm policies. Market Control and Interventions | Markov | ˚ Interaction ˚ Influence of historical events | 'What if' scenarios | 50 years |

*(Continued)*

**Table 1.** (Continued)

| Author | Title | IC topic-of-interest | Model | Aspects of complexity | Optimization | Time frame |
|---|---|---|---|---|---|---|
| Loyo et al. 2013 [57] | From model to action: using a system dynamics model of chronic disease risks to align community action. | Evaluating care management and interventions of chronic conditions. | SD | ˚ Dynamism ˚ Soft variables ˚ Interaction | *'What if'* scenarios | 30 years |
| Matta et al. 2007 [58] | Evaluating multiple performance measures across several dimensions at a multi-facility outpatient center | Performance measures evaluation | DES | ˚ Individualization ˚ Interference ˚ Interaction ˚ Dynamism | *'What if'*; *'How to'* scenarios | 1 working day |
| Milstein et al. 2010 [59] | Analyzing national health reform strategies with a dynamic simulation model. | National Health Reform Evaluation | SD | ˚ Dynamism ˚ Soft variables ˚ Interaction | *'What if'* scenarios | 25 years |
| Nianogo et al. 2018 [60] | Impact of Public Health Interventions on Obesity and Type 2 Diabetes Prevention: A Simulation Study. | Evaluation of Public health intervention | ABM | ˚ Individualization ˚ Simultaneity of events ˚ Influence of historical occurrence ˚ Interaction ˚ Emergence ˚ Intelligent Adaptation | *'What if'* scenarios | Adult life |
| Norouzzadeh et al. 2015 [61] | Simulation Modeling to Optimize Health Care Delivery in an Outpatient Clinic | Health facility operations simulation | DES | ˚ Individualization ˚ Interference ˚ Interaction ˚ Dynamism | *'What if'*; *'How to'* scenarios | 2 years |
| Oh et al. 2016 [62] | Use of a simulation-based decision support tool to improve emergency department throughput | Health facility operations simulation | DES | ˚ Individualization ˚ Interference ˚ Interaction ˚ Dynamism | *'What if'*; *'How to'* scenarios | 2.5 years |
| Rashwan et al. 2015 [63] | Modeling behavior of nurses in a clinical medical unit in a university hospital: Burnout implications | Planning Health force | SD | ˚ Dynamism ˚ Soft variables ˚ Interaction | *'What if'*; *'How to'* scenarios | 1 working day |
| Rejeb et al. 2018 [64] | Performance and cost evaluation of health information systems using micro-costing and discrete-event simulation. | Evaluation of Information System | DES | ˚ Individualization ˚ Influence of historical occurrence ˚ Interference ˚ Interaction ˚ Dynamism ˚ Simultaneity of events | *'What if'*; *'How to'* scenarios | 1 to 5 years |
| Sugiyama et al. 2017 [65] | Construction of a simulation model and evaluation of the effect of potential interventions on the incidence of diabetes and initiation of dialysis due to diabetic nephropathy in Japan. | Evaluating care management and interventions of chronic conditions. | SD | ˚ Dynamism ˚ Influence of historic occurrences ˚ Interaction | *'What if'* scenarios | 35 years |
| Vataire et al. 2014 [66] | Core discrete event simulation model for the evaluation of health care technologies in major depressive disorder. | Evaluating care management and interventions of chronic conditions. | DES | ˚ Individualization ˚ Influence of historical occurrence ˚ Interaction | *'What if'* scenarios | 1 to 5 years |

\* The model has been used in several projects and the time provided corresponds the one used most recently by Vidyanti et al. 2015 [67]

Event Simulations (DES), and two Agent-Based Models (ABM). Finally, one paper combined three techniques, adding a sixth, Hybrid models (HM). Supported by complementary literature, we describe each technique features and use the selected papers to exemplify their

**Table 2. Summary descriptions of simulation modeling techniques.**

| Models [complementary literature] | Strengths | Limitations | Estimation considerations |
|---|---|---|---|
| **Markov Models** [16, 73] | ■ Discrete or Continuous time<br>■ Easy calculation<br>■ Statistically valid<br>■ Inclusion of multiple data sources<br>■ Transitions can be time-dependent | ■ Aggregate transition rates cannot account for individual behavior.<br>■ Markovian property is a strong assumption.<br>■ Clearly defined states and transitions | ■ ODE: Transition probabilities determine the values in each state at each point in time<br>For estimation of Transition Matrix:<br>■ Maximum Likelihood Estimation (+Laplace)<br>■ Bootstrap approach<br>■ Maximum a posteriori |
| **System Dynamics** [14, 69, 74] | ■ Based on a conceptual model of the system, presented in a CLD & SFD.<br>■ Structure determines the performance and behavior of the system.<br>■ Better suited for continuous processes, where capturing information flow and feedback are important considerations.<br>■ Discrete or Continuous time (time steps can be short enough to be considered continuous)<br>■ More suitable for modeling whole systems | ■ Cannot include discrete changes in variables state.<br>■ Validity relies on usefulness, not statistical accuracy.<br>■ Population-based model. "Individualization" is only capable within the structure limits.<br>■ Sensibility analysis needs to account for possible trends or changing variables.<br>■ Sensitive to measurement errors. Aggregate diff eq tend to smooth fluctuations | ■ Continuous-time: Ordinary differential equations (ODE) for each variable value over time, defined by functions for inflows and outflows.<br>■ Euler<br>■ Runge-Kutta-Felberg method<br>■ Discrete-Time:<br>■ Difference equation |
| **Microsimulations** [70, 72, 75] | ■ Structured as a state transition model.<br>■ Agent driven. But the structure is important.<br>■ Stochastic estimation<br>■ Agents can be defined at multiple levels.<br>■ Good for modeling random or stochastic behavior, like the ones found in aggregate populations (patient groups) | ■ Because of computational and conceptual limitations, microsimulations results are routinely provided without measures of precision.<br>■ Microsimulation models are normally computing, data, and human-intensive<br>■ Difficult to validate | ■ The stochastic transitions between states are defined by functions including the individual factors of the agent moving through states.<br>■ In general, ODE is used (Similar calculation to SD, but from the agent's perspective) |
| **Discrete Event Simulation** [14, 73, 74] | ■ Process-centric. Described a clearly defined chronological process.<br>■ More suited when individual history is relevant for future events, or when queuing is a driver of performance.<br>■ Produces statistically valid representations of historical behavior.<br>■ Allows different cycle time lengths.<br>■ Discrete state, discrete time<br>■ Produces accurate and valid patient-level assessments of multiple interventions simultaneously, considering other important causal effects | ■ Needs a large amount of data and a specialized interpreter.<br>■ Computationally and human-intensive<br>■ Rigid in statistical validity, cannot include theories of qualitative relations.<br>■ More suitable as an assessment tool after a detailed risk prediction per patient | ■ODE with discrete states: discrete event system<br>■ There is a randomized sampling of time-to-event of future events, organized chronologically, that will determine the next action of the system. The list is rewritten after every event |
| **Agent-Based Simulations** [70, 75–78] | ■ Studies complex social phenomena<br>■ Describes system from the perspective of its constituent units.<br>■ Agents can be defined at multiple levels.<br>■ Technically simple<br>■ Validation and calibration are based on replicating real behavior.<br>■ Initial values are important | ■ Constructed under fully simulated conditions, some might discount the value of findings.<br>■ Due to uncertainty in data inputs and modeling process, ABM does not predict well, results are better interpreted qualitatively.<br>■ Computationally intensive; Each agent needs a definition and if stochasticity is used, computer usage is intensified | ■ Discrete model; estimation over simulation<br>■ Agents have a set of rules defining their behavior, and they are simulated to interact in an environment.<br>■ The effect is measured throughout the simulation |

implementation in integrated care. Table 2 summarizes the simulation techniques in terms of strengths, limitations, and estimation considerations and provides references for complementary literature.

**Markov models.** Markov models are state transition models. They have clearly defined, exclusive states, and transitions between states are defined as quantities per cycle. States cannot happen simultaneously for the same agent and transitions from one state to another depend

only on the current state (Markovian property). Time can be continuous or discrete, but in the case of this review, both papers use discrete time. Markov models can define transition probabilities differently for each time step, allowing the inclusion of trend factors, and together with 'tunnel states' (states with no possibility of remaining in the said state in time) time-depending dynamism and partial influence of historic events are enabled. Laurence et al. [53] explore the complexity of state transitions by constructing a model comprised of four separate parts (demand, supply, productivity, and training) of the system determining the health force gap, a common topic on integrated care initiatives. The demand and training parts of the model define partial outcomes dependent on several variables. These outcomes are then used in a second stage for the supply and productivity parts of the model, resulting in further partial outcomes. The third stage studies the main outcome (workforce gap) influenced by the outcomes of the previous stages. The structure enables the inclusion of mediated relationships between the initial variables, their interaction with partial outcomes, and the main outcome. The SimSmoke simulation, presented in Levy et al. [56] was developed in the early 2000s to estimate the smoking population and the effects of possible lifestyle interventions. The model distinguishes a population by age and gender evolving through birth and death rates. The population is further divided into never, current, and former smokers. By differentiating models for different strata of the population and including tunnel states, the author can represent the influence of historical events, having portions of the population 'jumping' to the next model when an event happens.

**System dynamics.**  The objective of system dynamics is to capture all determinant variables, causal pathways, and feedback loops of the system to be analyzed [48]. In SD structure determines performance, and the primarily goal is to evaluate the effect of an intervention over the qualitative nature of system performance (e.g. growth function, overshoot and collapse, oscillations, chaotic response, etc.) [27, 68]. To conceptualize the structure, relevant elements and the direction and nature of their inter-relations must be known. This information is extracted from the system's stakeholders underlying knowledge of the way the system operates [37, 59]. This way, Homer et al. [48] and Loyo et al. [57] integrate the most important risk factors of several chronic diseases in a single model. The model calculates the expected prevalence and indirect cost effect of these diseases in the population. Milstein et al. [59] include all relevant causal pathways related to health reform policies in the US. Kang et al. [51] and Sugiyama et al. [65] use the same approach to model the care of chronic kidney disease and the effect of interventions over diabetes and dialysis. The inclusion of all known determinants and causal pathways is complemented with the possibility to include "soft" variables, enabling the exploration of aspects of a system behavior particularly relevant to integrate care such as "Gaming", "Extrinsic motivation" [37], "Insurance complexity", "Care coordination" [59], "Staff resistance to new policies" or "Workload pressure" [63]. This flexibility is essential to capture the influence of important variables but limits the statistical validity of the results [69]. Loyo et al. [57] undermine this limitation stating that 'community decisions need to be made even though the data are disparate and incomplete'.

The model structure is represented in a causal loop diagram. There is a special focus on capturing the correct feedback loops affecting the system behavior. Feedback loops are what makes the system dynamic, by influencing the nature of the relationship between variables as the system progresses.

In the area of chronic disease management, Jones et al. [49] use causal loops to model the states of the disease itself, understanding that a key determinant in diabetes care is the reinforcement loop generated by the relation between the disease diagnosis behavior and detrimental consequences. When assessing the effect of a new nicotine product, Hill et al. [47] integrate the feedback effect of 'normality of smoking' to predict smoking initiation and

quitting rates, while Alonge et al. [37] introduce the negative feedback loop of gaming to understand the failure of a pay for performance incentive scheme in Afghanistan.

The structure of the system is transformed into a stock-and-flow-diagram, defining the nature of the elements presented in the causal loop diagram. Stocks (elements that accumulate value) and variables that influence flows (functions that determine the growth or decline of the value in stock) are differentiated. Functions are established for flows and initial quantities are assigned to stocks, so that differential equations can be used to determine the values in the stocks over time. Ansah et al. [38] uses this structure to set up the labor market for long term care, and uses a deterministic approach to study the effect of policies to reduce unwanted market disturbances. de Andrade et al. [41] use system dynamics to represent the different stages of the maturing process related to the management of a myocardial infarction case in a hospital environment. This type of structure is known as "Aging Chains" and is useful to gather information about how long the modeled entity stays in each stage and test delays-improving policies.

**Microsimulations.** As Markov Models, microsimulations are also state transition models, but they describe the population dynamics at individual levels and can be used to describe interactions between policies and individual decision-making units [70]. As state transition models, they are structured by clearly defined states. Transitions between states are generated by stochastic processes out of the parametrization of transition evidence, differentiating from the rational responses following an objective of Agent-Based Models or the time to event of Discrete Event Simulations [70, 71]. Even though the structure is similar to Markov models, they do not share some of the limitations. Besides the interaction of relevant variables, the individual approach adds the possibility of including 'tracking variables', to account for historical occurrences. Modeling the complexity of factors contributing to health care cost is the key objective of the "Future elderly model" created by Goldman et al. [45]. In said model, individualization and influence of historical occurrences allows for the inclusion of a multidimensional characterization of health status accounting for risk factors such as smoking, weight, age and education, along with lagged health and financial states. In their dynamic form, microsimulation models allow individuals to change their characteristics due to endogenous factors within the model [72]. In this sense, they are more suitable for modeling processes and large population dynamics, like the model Lay-Yee et al. [54] uses for estimating child health utilization. The authors modeled a child with a set of attributes as a starting point. Using equations derived from statistical analysis of real longitudinal data, they set the rules for the individual in the system and stochastically simulate changes in status over time. In other words, the model generates a set of diverse synthetic health histories for a starting sample of children. Then it uses the simulated sample as a counterfactual for estimation including the effect of interventions.

**Discrete event simulation.** Discrete event simulation is a process-centric simulation methodology that describes a chronological sequence of events affecting an entity. The entity (e.g., patients) carries its information, individualizing the type of relationship with each event. Vataire et al. [66] and Cooper et al. [40] use this characteristic for individualizing treatments for major depressive disorder, and to realistically assess the response to the prescription of prevention drugs for cardiovascular disease, respectively. All occurrences are registered in the entity's information, enabling the influence of historical events in future outcomes [74]. Getsios et al. [44] use this feature to model the effect of smoking cessation attempts in tobacco-related outcomes.

Events are listed in order after random sampling over the parametrization of time-to-event evidence, rewriting the list after each occurrence. Events have their own associated time that passes when the event occurs, hence DES is best suited to model discrete processes. As events

have different duration, the cycle lengths are not necessarily equal. Several authors [42, 46, 55, 62] find this structure convenient for modeling the care pathway of a health facility. The timing structure of a DES model allows the assessment of multiple and competing risks, as they will be organized in the future events list by time-to-event [79], with no immediate restriction for two events to happen simultaneously [73]. Kotiadis [52] and Norouzzadeh et al. [61] take advantage of this characteristic to model different times for referrals depending on medical factors while tracing key indicators in the system. DES also allows for the status of variables in the system to affect the nature of the relationships of an individual with the rest of the system. Günal et al. [46], Oh et al. [62] and Comans [39] uses the interference feature to evaluate the queues and backlogs at different stages of the patient pathway, understanding waiting time as a change in the manner a patient interacts with a provider, given the providers' status (e.g., 'Occupied'). By fixing the maximum waiting time allowed in concordance with national guidelines, the authors can assess the requirements in the rest of the system to reach this goal.

As Microsimulations, Discrete Event Simulations aim at producing statistically valid estimations out of the documented behavior of a system. This rigidity poses an important trade-off compared to other techniques as it needs detailed, well-defined processes, accurate historical data, and high intellectual, computer, and data management capabilities. Standfield et al. [73] conclude that if individualization or interference is not an important driver of the performance of the system, including these characteristics would be an unnecessary over-specification and unlikely to be informative to decision-makers.

**Agent-based models.**   Agent-based models focus on the activities of the agents composing the system. Each agent is individually defined with a set of rules and an objective, that may be described from heuristics to the optimization of a utility function. Kalton et al. [50] use this technique to model how mental patients engage with medical and social ecosystems while studying the effect of coordination capabilities. The individualization allows the agents to be influenced by their history and external variables. At the same time, agency focus allows the technique to capture emergent population phenomena [76].

The system is modeled in a simulated space, adding the possibility to include spatial variables. Nianogo et al. [60] exploit these characteristics when understanding the dynamics of the diabetes population in L.A, USA. The 'Virtual Los Angeles Obesity' model simulates a cohort of patients with different characteristics that interact differently with different environments. By assigning rules for the relations with the environment, the model seeks to describe the trends in obesity and diabetes out of the behavior of the agents, and at the same time test interventions by changing the environmental conditions or characteristics of said agents.

Agent-based models also allow for the inclusion of random factors to consider the bounded rationality that is present in agents' behavior. Finally, as agents can be affected by spatial or other types of determinants, and because the rules commanding agent's behavior can be set as thresholds, endogenous and time-dependent feedback loops are also possible. In advanced models, agents can evolve and learn with methods like neural networks and other forms of machine learning [29, 77, 78].

As with System Dynamics, authors use proxies and expert opinions when hard evidence is not available [46]. This flexibility makes them appropriate to test behavioral theories and understand complex population phenomena. On the other hand, statistical validity is not usually the first concern in either technique, where the usefulness of the assessment is more important.

**Hybrid simulations.**   Hybrid simulations can combine the strengths of two or more models. Gao et al. [43] developed a tripartite model combining System Dynamics, Agent-Based Models, and Discrete Event Simulations. He uses a previously developed System Dynamics model to understand the progression of diabetes up until the early stage of renal disease. As described by Jones et al. [49], the model properly describes diabetes progression by including

**Table 3. Complexity aspects enabled per simulation modeling technique.**

|  | Markov Model | System dynamics | Micro-Simulations | Discrete Event Simulation | Agent-Based Models |
|---|---|---|---|---|---|
| Individualization | X | X | ✓ | ✓ | ✓ |
| Dynamism | ✓[1] | ✓ | ✓ | ✓ | ✓ |
| Interaction | ✓ | ✓ | ✓ | ✓ | ✓ |
| Interference | X | X | X | ✓ | ✓ |
| Intelligent Adaptation | X | X | X | X | ✓ |
| Soft variables | X | ✓ | X | X | ✓ |
| Simultaneity of events | X | X | X | ✓ | ✓ |
| Influence of historical occurrences | X[2] | ✓ | ✓ | ✓ | ✓ |
| Emergence | X | X | X | X | ✓ |

[1] The technique can incorporate dynamic changes over time, but not endogenous feedback loops.

[2] Even though the 'Markovian Property' defines that transition probabilities will depend only on the current state and not on previous states thus eliminating the possibility of having 'Memory', researchers can overcome this by incorporating tunnel states and parallel models.

key feedback loops. Constructing from this model, Gao et al. [43] include two different types of hybrid relationships. First, there is an upstream-downstream relation between the original model and an Agent-Based Model for the populations that flows into a particular state (diabetes) to become individualized agents. The ABM model can study the incidence of a complication (early-stage renal disease) by simulating key behaviors in the development of the disease. In parallel, the second hybrid relation integrates DES for monitoring the different status of the patients and tracks the evolution of healthcare processes and resource availability and usage.

## 3.3 Complexity

To understand and compare the representation of complexity in simulation models we first compiled 13 distinct features of complex systems identified by Randall [80] and Wilenksy & Rand [81]: Undetermined or fuzzy boundaries, the possibility of being open, possibility of having nested sub subsystems, dynamism in the network of relationships with different scales of interconnectivity, emergent phenomena, nonlinear relationships, feedback loops, leverage points, memory/path dependence, sensitivity to initial conditions, robustness, diversity and heterogeneity, interconnectedness and interactions. Building from the previous section, we identified the characteristics of the described modeling techniques that can represent features of complex systems specifically related to relationships between system components. The modeled complexities were classified into one framework with definitions that could be applicable across methodologies. The exercise resulted in nine aspects of complex relations that can be represented with simulation models. We present the nine aspects of complex relations together with the characteristics in each discipline to represent them. In parenthesis, we show the number of papers modeling each complexity. Among the complexities identified, four are non-linearities (1 to 4), and they were the most commonly modeled. Table 3 summarizes the aspects of complexity enabled in each modeling technique.

1. Interactions (30/30): We understand this complexity as the dependence of the causal effect of one component (A) to another (B) on the effect of (C) over (A). i.e. mediated effects. For MM, SD, and MS interactions are embedded in the state transitions–chain structure. For DES and ABM, interactions between components are stored in their individualized information and will affect their effect on other components [70, 73, 75]. Homer's [69] model of policies aimed at chronic conditions presents a good example of interaction. The policies in question affect a risky behavior, which in time affects the status of the disease, which in

time affects healthcare provision. By interacting, each component affects the final outcome according to its particular characteristics and those of the previous component.

2. Dynamism (23/30): Dynamism represents the circular causality of a system. If component (A) changes the nature of its relations in the system as the system progresses, then we say the system presents dynamism. Besides the dynamics produced by the passing of time, relations can be influenced by the changing conditions of any other component, producing endogenous feedback loops. In methods where estimation correspond to ordinary differential equations, the value of component (A) will be determined by a function of the state of other components (B, C) [74]. For MM the other components (B, C) can only be time, hence no endogenous feedback loops are possible [73]. For ABM, conditions ruling the behavior of agents can change depending on other components of the system or time as programmed by the modeler [74]. In Alonge's [37] model for a pay for performance incentive scheme, dynamism is clear when understanding the effect of 'volume of service' over the reduction in 'quality' and the increase of 'revenue', which in time affect the 'volume of service' downwards and upwards respectively.

3. Interference (10/30): We understand interference as the dependence of the causal effect of one component (A) to another (B) on the effect of a third component (C) over (B). i.e queueing. DES handles interference by given the components of the system mutable states. The particular state will affect the relationship with other components, and at the same time mutations between states are triggered by these relations. Similarly, ABM can define different behaviors of its agents depending on current or past relations with the rest of the system [70, 75]. The best example of interference is the change from available to occupied of rooms modeled by Günal [46]. Because a patient is occupying a room, other patients have to change their behavior to that room and wait.

4. (Intelligent) Adaptation (2/30): Adaptation is the ability of a component to change the nature of its behavior to contingency happening in the system. This ability presumes the intelligence of components to make decisions. ABM can integrate this complexity when specifying agents' behavior not only as a function of other system components but also as conditions and operations in said function such as 'ifs' and optimization [14, 75]. For example, in Kalton's model [50] patients can make up to 40 decisions based on logic and preferences developed during their life process, care experience and health status. Decisions include taking their medicine, looking for employment, starting to abuse substances, etc.

5. Soft variables (9/30): Refers to the possibility of incorporating simplified proxies for difficult-to-measure variables. Allows the inclusion of behavioral and qualitative relations. The possibility of using soft variables in ABM [82] and SD [76] responds to each methodology obtaining outputs focusing on agents' behavior and system structure respectively, instead of mathematical correctness to represent phenomena. A good representation of a soft variable is "Workload pressure" modeled by Rashawn et al. [63] as the ratio between the actual nurse-to-patient ratio and the standard nurse-to-patient ratio.

6. Individualization (17/30): Integrates the possibility of including individual-level characteristics. Comprehends the complex system features of heterogeneity and diversity. DES and MS use a sample of individual units, each with a unique set of attributes [73, 75]. ABM can program each agent with different characteristics [82]. Individualization is notable in the model by Lay-yee et al. [54], where data is granular at patient level, with variables such as gender, ethnicity and housing status. Each of these variables affects the subject's number of doctor visits, reading ability and conduct problems.

7. Simultaneity of events (5/30): Possibility of two or more events happening in parallel for the same component of the system. The concept is related to the possibility of having nested systems within a complex system. Modelers of ABM can create parallel behavior rules for the same component. Similarly, events triggering a particular state can overlap in DES, creating parallel situations for the same component. A clear example is the case of Parkison disease treatmeant as modeled by Lebcir et al. [55], where one or a combination of the diferent treatment schemes are possible for distinct patients. When a combination is chosen, the treatment sections of the model happen in parallel.

8. Historical occurrences / Memory (18/30): Also known as hysteresis, the concept includes path dependence. It refers to the influence of past states on the nature of the relationships of the current state. In methods that allow individualization, events can be stored in the individual's characteristics. For SD, the influence of events is stored in the stocks. A good example is the model by Vataire et al. [66], where the number of previous depression events updates the model attributes.

9. Emergence (2/30): Characteristics of a system to develop new behaviors, different from those of the sum of its parts. ABM enables this characteristic by allowing agents to interact freely, only following the programmed behavior [82]. For example, in Nianogo's model for policies to treat population obesity [60], researches realize that their agents would change non objective behaviors because of the interventions, making them ineffective. Also, agents would quickly go back to the undesirable behavior after the intervention was finished (in despite of the intervention objective), diminishing the long-term effect.

## 3.4 Optimization capabilities

All simulation modeling techniques used '*what if*?' scenarios, defined as to gain information about the performance of the system (or parts of the system) when simulating the change of a variable from its original value, while using as counterfactual the baseline model. Fourteen (out of 30) articles complemented the assessment with '*how to*?' scenarios, defined as fixing a variable's value as a goal and focusing on how the other variables change from the baseline values to meet this condition.

## 3.5 Long term assessment

The studies had different time lengths in their assessment. While some papers had a closer look at the activities on a working day (3/30), the majority had assessments of at least 5 years (21/30). The mean number of years in the assessments was 18 years (standard deviation 20). Lifelong simulations (2/30) were considered as 60 years and working hours of a working day as 10 hours.

## 4. Discussion

We have characterized the use of simulation models for IHS performance assessment. First, by exposing topics of interest to IHS that can be modeled, and the techniques to model them. Second, by exposing how these techniques can implement system thinking in said topics of interest, while enabling features befitting of integrated care performance assessment.

To characterize the ability of the reviewed simulation models to implement system thinking, we have created a common framework with 9 complexity features enabled differently across modeling technique. These complexity features allow for the correct understanding of

causality paths in a system's performance. For integrated care, this means enabling accurate accountability for system components and consequently, creates a better position to guide system improvement. Accurate accountability is necessary for value-based care, and especially value-based payment schemes, two key elements of integrated care initiatives [4, 11]. Furthermore, disentangling the complex relations between system components is the key to deal with comorbidities, identifying consumed resources, and implementing ad-hoc interventions [11]. While accurately representing the complex relations of the system is essential for the model structure, simulation models can optimize interventions by testing 'what if?' & 'how to?' scenarios. These scenarios simulate changes (or fix values, respectively) anywhere in the system and compare it to a baseline value of system performance. By doing so, SM provides an easy way to compare the value of multiple interventions, understand the value of each component and identify bottlenecks and other deficiencies in the system. At the same time, the term of assessment is manageable in function of the objective of the study. Short and long-term interventions aimed at improving efficiency, changing health behavior, and preventive care are an important part of the toolbox of IHS, and the possibility of assessing them and optimize their implementation in the correct time frame is expected when in pursuit of the Triple Aim [3].

The application areas identified in the review were in line with the findings of previous work focused on characterizing applications areas of simulation modeling in healthcare [18]. Likewise, the simulation techniques covered in this work are the most used and studied in literature. Markov Models are the simplest among simulation models, because of relatively low computer, human, and data needs. It is the preferred methodology when assessing situations with low complexity. System Dynamics models add the possibility of including feedback effects and soft variables with a population perspective, characteristics that make it more prevalent in the "Policy and Strategy" area, a realization in line with results of extensive reviews aimed at linking simulation methods and healthcare areas of application [22, 25]. Microsimulations and Discrete event simulations extend the complexity into individual-level assessments, which in place enables the influence of past events. The main difference between the two is that Discrete Event Simulations add the possibility of including interference. This characteristic makes it more suitable to understand health processes that require queuing, a common feature in the topic of "Health Resource Management". Furthermore, several authors coincide in that Discrete Event Simulation is the most common technique for evaluating the operation management of care facilities [19, 22, 25]. Agent-Based Models understand the behavior of the system out of the behavior of its agents. This simple definition allows the study of complex phenomena with a relatively simple technical construction. The technique can include all the described complexities, but the fact that works in an entirely simulated environment diminish the validity of its results.

A common characteristic of all the simulation modeling techniques is the inclusion of data from multiple sources and the possibility of a probabilistic estimation. Twenty out of the 27 papers performed a probabilistic sensitivity analysis, either with Monte Carlo simulations or other. A probabilistic estimation is not included as an aspect of complexity as we don't consider uncertainty to be unique to complex systems, and for the same reason, Monte Carlo simulations are not included as a SM technique to assess complex systems. However, the possibility to include probabilistic estimations allows the inclusion of uncertain evidence, which is essential for the comprehensiveness of the models. Validation is key for the usefulness of the simulation results. Described in detail elsewhere [29], typically, a five steps approach is used in SM, comprising: Face validity, internal validity, cross validity, external validity, and predictive validity.

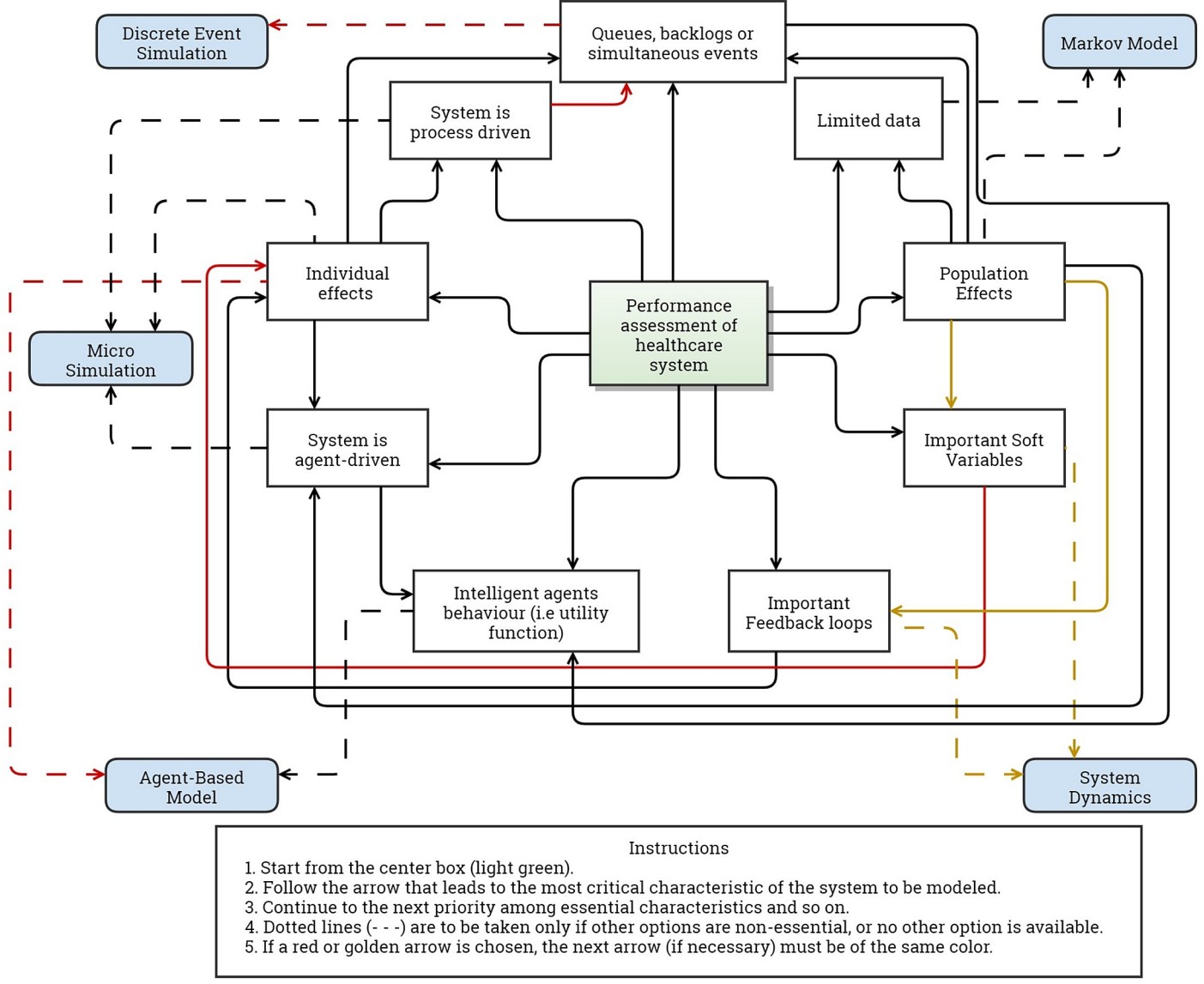

**Fig 2. Visual aid to select simulation modeling technique.**

## Model selection

A system is most appropriately modeled by the technique that allows the inclusion of the most important characteristics of said system. The selection of the most appropriate simulation modeling technique to assess performance must consider the characteristics of the system and the capabilities of each technique. It is important that only essential characteristics are considered so there is not an over-specification that hinders the analysis. In this line, identifying and prioritizing the complexities that rule the system to be modeled will help evaluators in selecting the most appropriate simulation model. Using our framework for complexity for this purpose, we created a conceptual map (Fig 2) that aids evaluators in selecting a simulation model to produce an accurate assessment of situations where complex relations are important. The tool is a summary of the results and characterization presented in this paper. The first step is to identify the most important complexity of the system to be modeled. Following a few key

questions, the tool points to the technique with fewer inputs and technical difficulties that is appropriate to model said system.

To help readers navigate the tool, we use the evaluation of a pay for performance incentive scheme by Alonge et al. [37] as an example. We start by assuming that the most important characteristics of the issue are (1.) the feedback loops that performance bonuses generate over the revenue and quality of services and (2.) the effect of "Gaming" (a soft variable) of the staff over this new payment scheme. Starting from the center and navigating through the figure we could go to either "Important feedback loops" or "Soft variables" and if individual effects are not considered essential, the tool takes us to System Dynamics—that is the approach used by the author. Another example is the evaluation of interventions for reducing waiting time in a health facility. Queues and backlogs are assumed the most important characteristic. If we consider non-essential the intelligent behavior of the agents, then the tool points to Discrete Event Simulation. Otherwise, an Agent-Based Model would be the most appropriate.

Sometimes the complexities of a system cannot be ranked according to their importance. If this is the case, evaluators should repeat the exercise starting from all the identified complexities as if each were the most important one. If the different runs result in different modeling techniques, a hybrid model is to be considered. This is the case for the paper by Gao et al. [43]. In this case the authors seek to model three elements of diabetes care. First, diabetes progression at the population level, with feedback loops being the most important complexity. Selecting important feedback loops in the figure takes you directly to System Dynamics (when individualization is not important). Second, disease complication, where individualization of risk factors and healthy behavior is crucial. After individual effects, the figure passes through agent behavior towards Agent-Based Models. Finally, the authors study the status of every patient to track the use of resources. In this case, individualization is the priority complexity, but as agent behavior is not important for this element, the user will lean in favor of simultaneity of events, arriving at Discrete Event Simulation. As selected by the authors, the tool guides each situation following the characteristics and prioritization of complexities to the appropriate modeling technique.

## Limitations

By focusing only on simulation modeling, the review overlooks many analytical methods to assess complex systems. Several authors have described other analytical methods for studying different aspects of complexity in health systems, including network analysis, marginal structural models, queuing theory, Petri nets [22], and artificial intelligence [83]. Previous work by Jun et al. [22] characterizes and compares a wider set of modeling methods. However, it does not consider the distinctive characteristics of the system to be modeled or describe how do they apply system thinking. Our review focuses solely on simulation models because of the advantages they present in the assessment of integrated care systems. Network Analysis provides an assessment of the structure of the (complex) relations in a system but does not consider causal pathways. Marginal structural models and queuing theory are useful to represent time-dependent covariates and interference (as defined in this paper) respectively, but they are limited to these capacities. SM and Artificial intelligence methods, such as Machine Learning, differ in that the latter constructs a model from patterns in the data, while SM constructs from the structure of the system and then populates the model with data. Besides making the estimations more comprehensible, this characteristic of SM allows policymakers to test structure changing interventions, such as the ones in integrated care. In any case, the mentioned analytical approaches are complementary to SM, as they can provide the necessary inputs to build and populate the simulation model. Comparisons between different analytical methods,

understanding their capacities to represent complex system characteristics, is scarce and should be further assessed in future research.

It is important to highlight that the approach to find IHS topics-of-interest is not the extent of subjectivity, as there are multiple definitions for integrated care [3]. In this sense, it is probable to encounter multiple other IHS topics-of-interest that can be successfully modeled with SM techniques. In the same line, our selection criteria focused on finding papers that allowed us to understand the implementation of a complex system perspective, criteria that resulted in fewer reviewed papers than previous literature linking simulation modeling and healthcare performance assessment. Nevertheless, we are confident that the selection of papers in the review together with the complementary literature used, allowed us to accurately characterize the field of simulation modeling in their ability to use system thinking in integrated healthcare.

Finally, clarify that our work does not provide an in-depth description of the different simulation modeling techniques. We acknowledge that such a task would be impossible to undertake with our study design. Instead, we provide readers with an introduction to the identified simulation modeling techniques and highlight the characteristics that allow them to implement system thinking. We encourage readers that find a solution in this work to the challenges they encounter when assessing the performance of a complex health system to learn in detail the technique that our paper has pointed towards. For this purpose, we recommend starting with the complementary literature that we include for each technique in Table 2.

## 5. Conclusion

Simulation modeling techniques can use system thinking and evaluate performance emphasizing the complex relations between system components, in topics of relevance for integrated healthcare systems. By using simulation models to complement the performance assessment of integrated health systems, managers can correctly attribute causality to system components, optimize interventions, and create long term assessments. All these are important advantages over traditional assessment methods. Adding simulation models to the performance assessment tools at disposition of health authorities may be the key to understand the full value of integrated care. Selecting a simulation technique is facilitated when both the characteristics of the modeling techniques are understood, and the complexities ruling the system performance are identified and prioritized. To facilitate the use of the discipline, we consolidated complexity features of different modeling techniques into one framework and provide future performance evaluators with a visual aid to guide the selection of the most appropriate model for the assessment of complexity-enhanced systems, such as integrated healthcare.

## Supporting information

**S1 Checklist. PRISMA 2009 checklist.**
(PDF)

**S1 Data. Database containing result of systematic search after removal of duplicates.**
(CSV)

**S1 Table. Search terms for systematic search.**
(DOCX)

**S2 Table. List of simulation modeling techniques and integrated care topics-of-interest used in selection criteria.**
(DOCX)

**S3 Table. Data extraction sheet.**
(DOCX)

# Acknowledgments

Ph.D. (Candidate) Sophie Wang (SW) contributed as a second literature reviewer and quality assessor.

# Author Contributions

**Conceptualization:** Nicolas Larrain, Oliver Groene.

**Data curation:** Nicolas Larrain.

**Formal analysis:** Nicolas Larrain.

**Investigation:** Nicolas Larrain.

**Methodology:** Nicolas Larrain.

**Project administration:** Nicolas Larrain.

**Software:** Nicolas Larrain.

**Supervision:** Oliver Groene.

**Visualization:** Nicolas Larrain.

**Writing – original draft:** Nicolas Larrain.

**Writing – review & editing:** Nicolas Larrain, Oliver Groene.

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
