## [Decision Letter · Decision Letter 0]

22 Sep 2020

PONE-D-20-10819

A systematic review of simulation modeling to assess health system performance:

characterization of the field and visual aid to guide model selection.

PLOS ONE

Dear Dr. Larrain,

Thank you for submitting your manuscript to PLOS ONE. After careful consideration, we feel that it has merit but does not fully meet PLOS ONE’s publication criteria as it currently stands. Therefore, we invite you to submit a revised version of the manuscript that addresses the points raised during the review process.

Thank you for submitting your work to PLOS One. Although the reviewers and I see some merit in the study, there are major issue that need to be addressed before the article can be considered for publication, particularly related to the paper's contribution and theoretical positioning. The reviewers provide detailed suggestions for improvement, which I hope will guide you in revising your work.

We look forward to receiving your revised manuscript.

Kind regards,

Federica Angeli

Academic Editor

PLOS ONE

Journal Requirements:

We note that one or more of the authors are employed by a commercial company: OptiMedis AG.

2.1. Please provide an amended Funding Statement declaring this commercial affiliation, as well as a statement regarding the Role of Funders in your study. If the funding organization did not play a role in the study design, data collection and analysis, decision to publish, or preparation of the manuscript and only provided financial support in the form of authors' salaries and/or research materials, please review your statements relating to the author contributions, and ensure you have specifically and accurately indicated the role(s) that these authors had in your study. You can update author roles in the Author Contributions section of the online submission form.

2.2. Please also provide an updated Competing Interests Statement declaring this commercial affiliation along with any other relevant declarations relating to employment, consultancy, patents, products in development, or marketed products, etc.  

Reviewers' comments:

Reviewer's Responses to Questions

**Comments to the Author**

1. Is the manuscript technically sound, and do the data support the conclusions?

Reviewer #1: Partly

Reviewer #2: Partly

2. Has the statistical analysis been performed appropriately and rigorously? 

Reviewer #1: N/A

Reviewer #2: N/A

3. Have the authors made all data underlying the findings in their manuscript fully available?

Reviewer #1: Yes

Reviewer #2: Yes

4. Is the manuscript presented in an intelligible fashion and written in standard English?

Reviewer #1: Yes

Reviewer #2: Yes

5. Review Comments to the Author

Reviewer #1: Authors introduce the topics very clearly, exploring the whole health care performance context thoroughly. They illustrate the issue the paper should contribute to address: how to keep into account complex interactions amid factors those which impact triple aim. They also specify that their review wants to explore possible simulations for improving “what if and “how to” scenario processes, listing the major limits of actual models proposed by some institutions. Final aim – presenting a visual aid to select the most appropriate simulation - is clearly expressed.

Methods section details search strategy. Inclusion ad exclusion criteria are clean.

Authors discuss data extraction and analysis which they adopted. They carefully show the process they followed, focusing how each step contributes to an integrated frame which looks coherent with paper’s aims.

Results

The description of Areas of Assessment is readable as it connects its themes with the topics in a structured way. The PRISMA diagram contributes to clarity.

Table 1 classifies selected papers by described criteria.

The description of simulation modelling techniques starts classifying them by five categories and adding a sixth one which includes studies based on three or more models (hybrid). The section structure is coherent but it could have started explaining reasons behind authors’ choice. Even though it should be considered a minor issue, it would add value trying to adopt a specific classification for simulation models. In case criteria available do not fit due to specificity of authors aims, it could help to quickly explore the connection between this classification and those are often adopted in decision making under uncertainty (e.g see the book Kochenderfer, M.J., 2015, Decision Making Under Uncertainty (MIT Lincoln Laboratory Series) The MIT Press.).

The table offers a synthetic view which allows readers getting the big picture. Once again, as minor issue authors should have quoted the sources they based upon to state strengths and limitations of different types of models. It could be especially helpful for readers who do not have specific background.

The Complexity section is especially interesting and at the center of authors aims. Lack of complexity in evaluating health care performances is one of the issues that authors want to address, so this section is expected to be rigorous and original. The latter expectation is quite satisfied while it is not possible to evaluate the first. Authors list nine complexity features those which are present in the 27 simulation models they’d previously selected. These features sound impactful and relevant but author should quote studies to help readers see them in the broader frame of system modelling theories. It also should allow to better understand both the connection among these features and how they impact on estimating health care system performance. Some features look ambiguous if considered outside a frame of reference as in different field they are referred to different phenomena. E.g. dynamism in dynamic system theory could refer to different meanings ranging from the presence of state variables to the time-variant characteristic of the system itself or both. Again, authors connect Adaptation with intelligence as the ability to make decisions following specific rules. While it is a possible option, the definition of dynamism few rows above could bring someone wonder whether these rules must vary over time for a model showing dynamism.

In summary, while this section is relevant and innovative, authors should better explain references and help readers put these features in a unitary frame.

Discussion

The discussion starts exploring why a family of models can be helpful in modelling a specific system. While the intent is correct, this part sounds a little bit narrative. Maybe a more schematic description could help to stay connected with both the nine features and the five types of models described above. This part looks as it was written through some and partial examples, so the reader could wonder why other considerations were neglected by authors and especially why.

In the second and in the third part authors discuss the core of the paper: how to improve the choice of performance simulation model for evaluating health care performance in the late of the “triple aim”.

They begin explaining how different models can cope with different health care systems, then they explain how they applied these concepts to design the visual tools, showed in fig. 2.

The visual tools is exciting for its simplicity. Authors illustrate through one example how to apply it for choosing a simulation model which fits both your needs and constraints. Nonetheless it shows a major weakness: authors do not specify how to integrate its different components. E.g. if the system under scrutiny calls for more than one relevant feature should the user follow different connectors, probably ending up in more than one loop? In this case, should users integrate different models?

Authors should be more systematic discussing the tool which should be considered like a model itself, as it offers a way to choose simulation models through matching the features proposed by authors, as a result of their review, and needs coming from triple aim approach. More examples should be proposed to help readers understand how to use the tool when more than one feature is necessary.

Limitations are expressed clearly.

Conclusion suffers the weakness posed in the discussion section.

Summary

Positive

• Relevant goals and clear expression of the issues to address

• Impactful contents

• High quality review process

• Clear language

To improve

• Clarify classification criteria

• Better connect the nine features in a common frame

• Explore and explain how to use the visual tool when health system calls for multiple features and needs

Reviewer #2: While this paper is well written, and discusses a timely topic that in principle is worthy of academic investigation, I do not think that the paper should be accepted for publication. In my view, the paper provides an interesting overview of recent simulation modeling efforts in health systems (with a special emphasis on complexity, being an important characteristic of such systems). But it does not go beyond that, and indeed more is needed to justify publication in a scholarly journal.

First, the paper misses a scientific motivation. The justification for the paper is given in lines 70-72 and can be summarized by saying that simulation models have recently gained more recognition. But this in itself is no (scholarly) motivation for reviewing the related literature. A proper motivation would include: a discussion of why and to whom such a review would be beneficial, i.e., what is the aim of doing such a review? Who would use it, and to what aim? What goes wrong when the study (review) is not done? Also, a scientific motivation would include the identification of a clear knowledge gap: what other review efforts have been published in the academic literature? What is missing in those reviews, making this review necessary? Can the health domain learn from reviews of simulation models in other fields? Especially in light of various recently published review papers (14, 24, 65) it is of paramount importance that the authors justify the need for another review paper on this topic. Without such a clear motivation, the paper reads more like a (interesting) policy report, not a scholarly article.

Second, although the paper provides an interesting overview of relevant papers and the properties of the models described in them, it does not offer as much as I hoped for, in terms of deep reflection and discussion of subtle connections between the models. This can be due to the fact that an enormously wide net is cast across a variety of modeling approaches, each of which having been studied in thousands and thousands of academic papers. Describing such canonical and grand modeling traditions in so few words inescapably leads to gross simplifications. Although understandable, this does not do justice to the subtleties and complexities of each of these models, let alone the nuanced relations between them. Similarly, the applications of the discussed models in the healthcare domain are so diverse (see column IC Topic-of-interest in Table 1), that drawing commonalities and generic lessons from them is hardly possible. What can one learn from comparing the use of model A in context X with the use of model B in context Y? For this to lead to credible results, many more data points are needed. As a consequence, the paper results in a well-structured but unavoidably shallow discussion.

These two issues together make that this reviewer, having read the paper, feels that the paper never really explained what its added value to the academic literature would be (point 1) and that it is indeed difficult to identify key lessons learned from the material that is presented (point 2). In combination, this makes the paper rather unsuitable for publication in a scholarly journal, although I can imagine that with some extensions the manuscript could be made suitable as a report for practitioners and policy makers, especially those with limited knowledge of simulation models and a wish to have a systematic, practical guide to help them navigate such models in the healthcare domain.

6. PLOS authors have the option to publish the peer review history of their article (what does this mean?). If published, this will include your full peer review and any attached files.

Reviewer #1: **Yes: **Andrea Montefusco

Reviewer #2: **Yes: **Caspar Chorus

---

## [Author Response · Author response to Decision Letter 0]

19 Nov 2020

Dear reviewers and Academic Editor;

We deeply appreciate your constructive critique and valuable comments. We have extensively revised the manuscript to address each of the concerns.

Reviewer 1.

Comment 1: Authors introduce the topics very clearly, exploring the whole health care performance context thoroughly. They illustrate the issue the paper should contribute to address: how to keep into account complex interactions amid factors those which impact triple aim. They also specify that their review wants to explore possible simulations for improving “what if and “how to” scenario processes, listing the major limits of actual models proposed by some institutions. Final aim – presenting a visual aid to select the most appropriate simulation - is clearly expressed.

Methods section details search strategy. Inclusion ad exclusion criteria are clean.

Authors discuss data extraction and analysis which they adopt-ed. They carefully show the process they followed, focusing how each step contributes to an integrated frame which looks coherent with paper’s aims.

Response 1: Thank you very much for your positive assessment of our methodological approach.

Comment 2: Results

The description of Areas of Assessment is readable as it connects its themes with the topics in a structured way. The PRIS-MA diagram contributes to clarity.

Table 1 classifies selected papers by described criteria.

The description of simulation modelling techniques starts classifying them by five categories and adding a sixth one which includes studies based on three or more models (hybrid). The section structure is coherent but it could have started explaining reasons behind authors’ choice. Even though it should be considered a minor issue, it would add value trying to adopt a specific classification for simulation models. In case criteria available do not fit due to specificity of authors aims, it could help to quickly explore the connection between this classification and those are often adopted in decision making under uncertainty (e.g see the book Kochenderfer, M.J., 2015, Decision Making Under Uncertainty (MIT Lincoln Laboratory Series) The MIT Press.).

Response 2: The reviewer is correct in raising that the classification used for Simulation models and the structure used in the results section requires further explanation. As the reviewer mentioned, the reason is that criteria previously reported in the literature do not fit well the practical applications of our study, which we have now further specified. Furthermore, several reviews have been published with different approaches of classifying simulation models. Our structure choice purposely differentiates from these classifications in an effort to expose clearly the added value and practical implication of our paper. Nevertheless, we do agree that an explanation of our choice was necessary and was added in section 2.4, on lines 138 - 155, including the references that guided each item.

We thank the reviewer for the reference provided. In studying it we believe that its scope is more aligned with classifications responding to technical and mathematical issues (presented in a complexity progression typical of academic books) more than an exposé of usefulness for a particular area(Integrated care), like our work. We have however added a short mention to decision making under uncertainty on the Discussion section, lines 431-437 of the final manuscript.

Comment 3: The table offers a synthetic view which allows readers getting the big picture. Once again, as minor issue authors should have quoted the sources they based upon to state strengths and limitations of different types of models. It could be especially helpful for readers who do not have specific background.

Response 3: We thank the reviewer and agree with the assessment. The sources have been referenced in the table.

Comment 4: The Complexity section is especially interesting and at the center of authors aims. Lack of complexity in evaluating health care performances is one of the issues that authors want to address, so this section is expected to be rigorous and original. The latter expectation is quite satisfied while it is not possible to evaluate the first. Authors list nine complexity features those which are present in the 27 simulation models they’d previously selected. These features sound impactful and relevant but author should quote studies to help readers see them in the broader frame of system modelling theories.

Response 4: We thank the reviewer for the positive assessment of the originality of our study. In order to allow evaluation of the rigor employed, we have now added an expanded explanation of the common framework to evaluate complex relations between system components (section Results; 3.3 Complexity; lines 333-346). References for the terms and concepts in the different fields were provided. 

Comment 5: It also should allow to better understand both the connection among these features and how they impact on estimating health care system performance.

Response 5: We agree and have updated the explanations for better internalization of the concepts (lines 347-391).

Comment 6: Some features look ambiguous if considered outside a frame of reference as in different field they are referred to different phenomena. E.g. dynamism in dynamic system theory could refer to different meanings ranging from the presence of state variables to the time-variant characteristic of the system itself or both. Again, authors connect Adaptation with intelligence as the ability to make decisions following specific rules. While it is a possible option, the definition of dynamism few rows above could bring someone wonder whether these rules must vary over time for a model showing dynamism.

Response 6: We thank the reviewer for raising this issue. We added references to a common frame-work to consolidate the concepts from different fields. (lines 333-391)

Comment 7: In summary, while this section is relevant and innovative, authors should better explain references and help readers put these features in a unitary frame.

Response 7: We are particularly thankful for this comment as it motivated us to present and give central importance to the complexity framework we had developed.

Comment 8: Discussion

The discussion starts exploring why a family of models can be helpful in modelling a specific system. While the intent is correct, this part sounds a little bit narrative. Maybe a more schematic description could help to stay connected with both the nine features and the five types of models described above.

This part looks as it was written through some and partial examples, so the reader could wonder why other considerations were neglected by authors and especially why.

Response 8: Based on the reviewer´s comment we have now reorganized this section and changed the discussion with two main focuses. 1st to discuss the advantages for integrated care of implementing systems thinking, including a summary of the contribution of each modeling technique and the discipline in general in this regard.

2nd On the usefulness of our research and how to use the tool that summarizes our findings.

Comment 9: In the second and in the third part authors discuss the core of the paper: how to improve the choice of performance simulation model for evaluating health care performance in the late of the “triple aim”.

They begin explaining how different models can cope with different health care systems, then they explain how they applied these concepts to design the visual tools, showed in fig. 2.

The visual tools is exciting for its simplicity. Authors illustrate through one example how to apply it for choosing a simulation model which fits both your needs and constraints. Nonetheless it shows a major weakness: authors do not specify how to integrate its different components. E.g. if the system under scrutiny calls for more than one relevant feature should the user follow different connectors, probably ending up in more than one loop? In this case, should users integrate different models?

Authors should be more systematic discussing the tool which should be considered like a model itself, as it offers a way to choose simulation models through matching the features pro-posed by authors, as a result of their review, and needs coming from triple aim approach. More examples should be proposed to help readers understand how to use the tool when more than one feature is necessary.

Response 9: As we said before, we thank the reviewer for pushing us to give more emphasis to our selection tool. We have added a general rationale for use of the tool (Discussion, lines 451-456) and a third example that covers how to deal with a system with multiple parallel complexities (467-479). The key to the use of the selection tool is the identification and prioritization of the complexities that rule the system’s performance.

We acknowledge the limitation of the tool to integrate several complexities with a different priority. However, after the selection of the most important complexity to be modeled, the transit options will integrate the most com-mon second or third level complexities associated with systems ruled by the selected top complexity. We believe the practical utility of our tool to orient users towards an appropriate modeling approach justifies our approach

Comment 10: Limitations are expressed clearly. Conclusion suffers the weakness posed in the discussion section.

Response 10: Based on the reviewer's comments, we have updated our conclusion in line with the new structure of the discussion.

-------

Reviewer 2.

Comment 1: While this paper is well written, and discusses a timely topic that in principle is worthy of academic investigation, I do not think that the paper should be accepted for publication. In my view, the paper provides an interesting overview of recent simulation modeling efforts in health systems (with a special emphasis on complexity, being an important characteristic of such systems). But it does not go beyond that, and indeed more is needed to justify publication in a scholarly journal.

Response 1: We thank the reviewer for stating that we are providing an interesting overview of recent simulation modeling efforts in health systems, one of the objectives of our paper.

However, we kindly disagree that our paper does not go beyond that.

First, while we clearly state that previous reviews have been conducted in the field, some of them are quite outdated. Our review provides a thorough and up-to-date overview of current approaches in the field.

Secondly, as also pointed out by reviewer one we have presented an innovative approach to help users identify the most appropriate simulation technique, given a set of criteria. We believe that our presentation of this approach would indeed stimulate scholarly debate and may lead to further refinements of the method.

Comment 2: First, the paper misses a scientific motivation. The justification for the paper is given in lines 70-72 and can be summarized by saying that simulation models have recently gained more recognition. But this in itself is no (scholarly) motivation for reviewing the related literature. A proper motivation would include: a discussion of why and to whom such a review would be beneficial, i.e., what is the aim of doing such a review? Who would use it, and to what aim? What goes wrong when the study (review) is not done?

Response 2: We thank the reviewer for the reflection which we considered carefully. The assessment of the reviewer might be informed by the expectation of a typical PICO style presentation of the research questions, which however does not apply to our study subject. In order to respond to the points raised by the reviewer, we have re-written the Introduction to better state the scientific motivation(57-59), the knowledge gap(71-82.), our aim(83-86) and objectives and the audience of the review(86-88).

In short, the paper’s motivation relates to the lack of tools to assess integrated care as a dominant health system reform strategy. The identified problem is that system thinking is missing in integrated care performance assessment framework.

A solution to this problem is simulation modeling. The identified knowledge gap is the link between the complex system perspective of SM and the usefulness for integrated care performance assessment. We close this gap by reviewing papers that use simulation models with systems thinking for evaluations in topics relevant to integrated care systems and extracting the particular features that are useful for evaluating integrated care. We focus on the features for implementation of system thinking in a common framework and culminate our results in a tool for guiding model selection. We hope that the reviewer is satisfied with these improvements to the manuscript.

Comment 3: Also, a scientific motivation would include the identification of a clear knowledge gap: what other review efforts have been published in the academic literature? What is missing in those reviews, making this review necessary? Can the health domain learn from reviews of simulation models in other fields? Especially in light of various recently published review papers (14, 24, 65) it is of paramount importance that the authors justify the need for another review paper on this topic. Without such a clear motivation, the paper reads more like a (interesting) policy report, not a scholarly article.

Response 3: We agree with the reviewer that an articulation of the knowledge gap required strengthening. We added a paragraph referencing previous literature and stating clearly the knowledge gap that we are covering with the review. Lines 71-82.

Comment 4: Second, although the paper provides an interesting overview of relevant papers and the properties of the models described in them, it does not offer as much as I hoped for, in terms of deep reflection and discussion of subtle connections between the models. This can be due to the fact that an enormously wide net is cast across a variety of modeling approaches, each of which having been studied in thousands and thousands of academic papers. Describing such canonical and grand modeling traditions in so few words inescapably leads to gross simplifications. Although understandable, this does not do justice to the subtle-ties and complexities of each of these models, let alone the nuanced relations between them.

Response 4: We thank and acknowledge the limitation highlighted by the reviewer and have addressed it in the limitations section (lines 503-509).

Nevertheless, (as the added paragraph clarifies) we think that the provided simplifications don’t undermine the usefulness and academic value of our work. We acknowledge that an in-depth description of the identified models that would allow an analysis of the subtle connections between models would be impossible to undertake with our study design. Instead, we provide readers with an introduction to the identified simulation modeling techniques and highlight the characteristics that allow them to implement system thinking.

Furthermore, we provide a way forward by encouraging readers that find a solution in our work to the challenges they encounter when assessing the performance of a complex health system to learn in detail the technique that our paper has pointed towards. Such an approach, where an extensive literature/discipline is undertaken by a few practical applications to provide a practical way forward, is common in the academic literature and fundamental for bringing together different fields of expertise. An example is the paper by Behrendt & Groene 2016 (DOI: 10.1016/j.healthpol.2016.08.003)

Comment 5: Similarly, the applications of the discussed models in the healthcare domain are so diverse (see column IC Topic-of-interest in Table 1), that drawing commonalities and generic lessons from them is hardly possible. What can one learn from comparing the use of model A in context X with the use of model B in context Y? For this to lead to credible results, many more data points are needed. As a consequence, the paper results in a well-structured but unavoidably shallow discussion.

Response 5: In agreement with the reviewer comment, we too believe that is not possible to discern clear patterns between models and types of application in integrated healthcare with such a small sample size. However, two incipient patterns do emerge; (1.) System Dynamics for “Policy and strategy” and (2.) Discrete Event Simulations for “Health Resource Management”. This analysis is mention in the Discussion (lines 419-431).

This being said, even though finding a relation between the type of model and the different applications would be useful for integrated healthcare managers, a similar classification has already been made (Jun et al 2011) and escapes our main objective, that is to characterize simulation modeling focusing on the ability to implement system thinking. 

Instead, we provide the type of applications (section 3.1, line 168) as part of a comprehensive characterization of simulation modeling in integrated care, with the purpose of informing readers of the reach of the discipline in integrated care topics. The classification also serves the purpose of grouping complexities, and we think it will help readers to identify complexities in different settings. 

Comment 6: These two issues together make that this reviewer, having read the paper, feels that the paper never really explained what its added value to the academic literature would be (point 1) and that it is indeed difficult to identify key lessons learned from the material that is presented (point 2). In combination, this makes the paper rather unsuitable for publication in a scholarly journal, although I can imagine that with some extensions the manuscript could be made suitable as a report for practitioners and policy makers, especially those with limited knowledge of simulation models and a wish to have a systematic, practical guide to help them navigate such models in the healthcare domain.

Response 6: We thank the reviewer for the critique, which we have carefully considered and based on which we have substantially revised the manuscript.

With utmost respect, we provide a summarized answer to the two points raised by the reviewer.

Point 1: Exposing simulation models as a solution for the challenges of integrated healthcare systems performance assessment, focusing particularly on their ability to implement system thinking.

Point 2. A summary of key (to IHS) characteristics and complexity features in a common framework that helps in guiding the identification and prioritization of complexity, which in turn are useful to select an appropriate simulation model.

We strongly believe that our methodological approach adheres to rigorous standards and that our work goes far beyond the content expected for a policy brief or practical guide and indeed would inform scholarly debate.

-------

Editor:

Comments:

We note that one or more of the authors are employed by a commercial company: OptiMedis AG.

2.1. Please provide an amended Funding Statement declaring this commercial affiliation, as well as a statement regarding the Role of Funders in your study. If the funding organization did not play a role in the study design, data collection and analysis, decision to publish, or preparation of the manuscript and only provided financial support in the form of authors' salaries and/or research materials, please review your statements relating to the author contributions, and ensure you have specifically and accurately indicated the role(s) that these authors had in your study. You can update author roles in the Author Contributions section of the online submission form.

2.2. Please also provide an updated Competing Interests Statement declaring this commercial affiliation along with any other relevant declarations relating to employment, consultancy, patents, products in development, or marketed products, etc.

Response: 

We thank the Academic Editor for your comments and guidelines. In response, we are pleased to declare the following:

1. We have formatted the manuscript as stated in the templates.

2. We have provided an amended Funding Statement in the Cover letter as recommended

3. We have provided an amended Competing Interests Statement in the Cover Letter as recommended.

---

## [Decision Letter · Decision Letter 1]

4 Feb 2021

PONE-D-20-10819R1

A systematic review of simulation modeling to assess health system performance: Characterization of the field and visual aid to guide model selection.

PLOS ONE

Dear Dr. Larrain,

Thank you for submitting your manuscript to PLOS ONE. After careful consideration, we feel that it has merit but does not fully meet PLOS ONE’s publication criteria as it currently stands. Therefore, we invite you to submit a revised version of the manuscript that addresses the points raised during the review process.

We look forward to receiving your revised manuscript.

Kind regards,

Yong-Hong Kuo

Academic Editor

PLOS ONE

Additional Editor Comments (if provided):

The referees from the last round were invited to review this revision. One of them agreed and returned the review report. The reviewer still had some comments for the authors to address in their work. The other was not available to review for this round. Thus, I have gone through his/her comments and the revision. Below is my evaluation.

1. As the reviewers suggested, the academic value of the work shall be strengthened. Currently, it is unclear about the research questions and how the state-of-the-art is advanced by this work. The scientific motivation is still missing.

2. The analysis part shall be strengthened. What are the key messages resulting from the analysis? Also, it would be nice to have this article discussing the research trends and shedding light on future research directions.

3. There have been dozens of literature review / survey papers on simulation models of healthcare applications. The position of this paper is unclear. How is this paper different from the existing review papers?

4. The number of papers on healthcare simulation is tremendous. Currently, only 27 papers were analyzed. This coverage is much narrow than those covered by the existing review papers. I suggest the authors have a more comprehensive review of the studies, particularly the recent ones. To my knowledge, the below studies are relevant to this review work. However, the list below is not exhaustive and the authors shall identify further related studies:

• Abramovich, M. N., Hershey, J. C., Callies, B., Adalja, A. A., Tosh, P. K., & Toner, E. S. (2017). Hospital influenza pandemic stockpiling needs: a computer simulation. American journal of infection control, 45(3), 272-277.

• Chen, Y., Kuo, Y. H., Balasubramanian, H., & Wen, C. (2015, December). Using simulation to examine appointment overbooking schemes for a medical imaging center. In 2015 Winter Simulation Conference (WSC) (pp. 1307-1318). IEEE.

• Gul, M., & Guneri, A. F. (2015). A comprehensive review of emergency department simulation applications for normal and disaster conditions. Computers & Industrial Engineering, 83, 327-344.

• Kuo, Y. H. (2014). Integrating simulation with simulated annealing for scheduling physicians in an understaffed emergency department. HKIE transactions, 21(4), 253-261.

• Kuo, Y. H., Leung, J. M., Graham, C. A., Tsoi, K. K., & Meng, H. M. (2018). Using simulation to assess the impacts of the adoption of a fast-track system for hospital emergency services. Journal of Advanced Mechanical Design, Systems, and Manufacturing, 12(3), JAMDSM0073-JAMDSM0073.

• Kuo, Y. H., Rado, O., Lupia, B., Leung, J. M., & Graham, C. A. (2016). Improving the efficiency of a hospital emergency department: a simulation study with indirectly imputed service-time distributions. Flexible Services and Manufacturing Journal, 28(1-2), 120-147.

• Hu, X., Barnes, S., & Golden, B. (2018). Applying queueing theory to the study of emergency department operations: a survey and a discussion of comparable simulation studies. International transactions in operational research, 25(1), 7-49.

• Moeke, D., van de Geer, R., Koole, G., & Bekker, R. (2016). On the performance of small-scale living facilities in nursing homes: a simulation approach. Operations research for health care, 11, 20-34.

• Niessner, H., Rauner, M. S., & Gutjahr, W. J. (2018). A dynamic simulation–Optimization approach for managing mass casualty incidents. Operations research for health care, 17, 82-100.

• Ordu, M., Demir, E., Tofallis, C., & Gunal, M. M. (2020). A novel healthcare resource allocation decision support tool: A forecasting-simulation-optimization approach. Journal of the operational research society, 1-16.

• Roy, S. N., Shah, B. J., & Gajjar, H. (2020). Application of Simulation in Healthcare Service Operations: A Review and Research Agenda. ACM Transactions on Modeling and Computer Simulation (TOMACS), 31(1), 1-23.

• Roy, S., Prasanna Venkatesan, S., & Goh, M. (2020). Healthcare services: A systematic review of patient-centric logistics issues using simulation. Journal of the Operational Research Society, 1-23.

• Salleh, S., Thokala, P., Brennan, A., Hughes, R., & Booth, A. (2017). Simulation modelling in healthcare: an umbrella review of systematic literature reviews. PharmacoEconomics, 35(9), 937-949.

• Vanbrabant, L., Braekers, K., Ramaekers, K., & Van Nieuwenhuyse, I. (2019). Simulation of emergency department operations: A comprehensive review of KPIs and operational improvements. Computers & Industrial Engineering, 131, 356-381.

• Vanbrabant, L., Martin, N., Ramaekers, K., & Braekers, K. (2019). Quality of input data in emergency department simulations: framework and assessment techniques. Simulation Modelling Practice and Theory, 91, 83-101.

• Weissman, G. E., Crane-Droesch, A., Chivers, C., Luong, T., Hanish, A., Levy, M. Z., ... & Halpern, S. D. (2020). Locally informed simulation to predict hospital capacity needs during the COVID-19 pandemic. Annals of internal medicine, 173(1), 21-28.

• Yousefi, M., Yousefi, M., & Fogliatto, F. S. (2020). Simulation-based optimization methods applied in hospital emergency departments: A systematic review. Simulation, 96(10), 791-806.

• Zhang, C., Grandits, T., Härenstam, K. P., Hauge, J. B., & Meijer, S. (2018). A systematic literature review of simulation models for non-technical skill training in healthcare logistics. Advances in Simulation, 3(1), 1-16.

Based on the reviewer's and my one evaluations, I recommend major revision.

Hope that the authors shall find the comments constructive. The revision will go through a rigorous review process again. Unsuccessful revision can lead to rejection of the work.

Reviewers' comments:

Reviewer's Responses to Questions

**Comments to the Author**

1. If the authors have adequately addressed your comments raised in a previous round of review and you feel that this manuscript is now acceptable for publication, you may indicate that here to bypass the “Comments to the Author” section, enter your conflict of interest statement in the “Confidential to Editor” section, and submit your "Accept" recommendation.

Reviewer #1: (No Response)

2. Is the manuscript technically sound, and do the data support the conclusions?

Reviewer #1: Yes

3. Has the statistical analysis been performed appropriately and rigorously? 

Reviewer #1: N/A

4. Have the authors made all data underlying the findings in their manuscript fully available?

Reviewer #1: Yes

5. Is the manuscript presented in an intelligible fashion and written in standard English?

Reviewer #1: Yes

6. Review Comments to the Author

Reviewer #1: The following comments refer directly to those in the previous review and the authors’ answers. The actual review did not limit to verifying whether the authors have addressed issues but reconsidered the entire paper.

Comment 1

No comment

Comment 2

The authors profoundly modified section 2.4. While before, it explained the data analysis approach weakly, now it meets expectations. Authors fully reformulate sentences, making possible a better understanding of the paper aims too. The section specifies, step by step, the rationale behind every item’s choice. It allows the reader to navigate the paper with a clean schema. The connections between items and the research question are now clear. Though this section calls for attention, it progressively helps readers build the first representation of how items integrate each other. It results in a sort of “integrated variable space” which appears to be the real novelty of this paper.

Comment 3

The authors followed the suggestion and added the reference column in Table 1. Though it was a minor issue, expert readers can now connect with their knowledge while everyone could search sources directly and efficiently.

Comment 4

The authors rewrote section 3.3 completely and, now the reader can find proper references. It helps understand the rationale behind the nine aspects of complex relations and the way to model them. It is easy to go through authors’ statements, as the text and the table together provide a clean, integrated view.

Comment 5

While the authors analyzed complexity dimensions well, section 3.3 still misses why the nine dimensions impact health system performance. They go through that in session 4, Discussion. That is correct, as this way, they directly connect their framework with the research question and the final paper’s goal. It could help anticipate here straightforward examples of each complexity component's impacts on health care system performance. Although this could be considered a minor issue, nonetheless, it can improve the clarity much. Perhaps it is not easy to connect these concepts with health care system performance for those who have no in-depth knowledge of complex system theory and health care. Despite being a minor issue, this could limit the paper’s practical impact. Delivering examples in section 3.3 could increase the readability of section 4, as the readers would have examples in their minds.

Comment 6

See comment 4.

Comment 7

See comment 5. The authors hit the nail, though they could have delivered examples of impacts in this section too.

Comment 8-9

The authors provided the suggested schematic connection between the nine features and health care. The section is informative now, though it still lacks a synthesis. Maybe its presence – A draw? Flow diagramm? - could help readers generalizing the application of the frame. Although the authors commented on the examples more in-depth than in the first submission, maybe the readers could find it challenging to apply the tools to a specific case.

With their thorough revision, the authors improved the first and the second parts of the paper. The final sections get better after the revision, but it requires further work to keep the previous sections' promises.

While reading it, it is hard to focus on the research question, as authors still do not explicitly connect their results’ discussion with keywords like triple aims and system performances, for just quickly quote them in the Limitations and Conclusion sections.

Conclusion and Limitation must exploit the potential that is now shown by the previous sections. At the moment, they seem to be far weaker than the rest of the paper.

7. PLOS authors have the option to publish the peer review history of their article (what does this mean?). If published, this will include your full peer review and any attached files.

Reviewer #1: **Yes: **Andrea Montefusco

---

## [Author Response · Author response to Decision Letter 1]

20 Mar 2021

Dear reviewers and Academic Editor.

We are deeply thankful for your revisions, valuable comments, and additional clarifications. We have extensively revised the manuscript to address each of your concerns. We have taken the liberty of organizing your comments in the following list, only including the second revision and responses organized by reviewer and commentary. We recommend attending the attached "Response to reviewers" letter, that provides both revisions and responses, for a better assessment.

Reviewer 1:

1. Comment 1:

2nd revision: No comment

Response 2nd revision: -

2. Comment 2:

2nd revision: The authors profoundly modified section 2.4. While before, it explained the data analysis approach weakly, now it meets expectations. Authors fully reformulate sentences, making possible a better understanding of the paper aims too. The section specifies, step by step, the rationale behind every item’s choice. It allows the reader to navigate the paper with a clean schema. The connections between items and the research question are now clear. Though this section calls for attention, it progressively helps readers build the first representation of how items integrate each other. It results in a sort of “integrated variable space” which appears to be the real novelty of this paper.

Response 2nd revision: We thank the reviewer for the positive comments.

3. Comment 3:

2nd revision: The authors followed the suggestion and added the reference column in Table 1. Though it was a minor issue, expert readers can now connect with their knowledge while everyone could search sources directly and efficiently.

Response 2nd revision: We thank the reviewer for his comments and contribution in this section.

4. Comment 4:

2nd revision: The authors rewrote section 3.3 completely and, now the reader can find proper references. It helps understand the rationale behind the nine aspects of complex relations and the way to model them. It is easy to go through authors’ statements, as the text and the table together provide a clean, integrated view.

Response 2nd revision: We thank the reviewer for his comments and contribution in this section.

5. Comment 5:

2nd revision: While the authors analyzed complexity dimensions well, section 3.3 still misses why the nine dimensions impact health system performance. They go through that in session 4, Discussion. That is correct, as this way, they directly connect their framework with the research question and the final paper’s goal. It could help anticipate here straightforward examples of each complexity component's impacts on health care system performance. Although this could be considered a minor issue, nonetheless, it can improve the clarity much. Perhaps it is not easy to connect these concepts with health care system performance for those who have no in-depth knowledge of complex system theory and health care. Despite being a minor issue, this could limit the paper’s practical impact. Delivering examples in section 3.3 could increase the readability of section 4, as the readers would have examples in their minds.

Response 2nd revision: We thank the observation by the reviewer, and we concur in his assessment. We included short and clear examples for each complexity feature in section 3.3, referencing to previously explained phenomena when possible.

6. Comment 6

2nd revision: -

Response 2nd revision: -

7. Comment 7:

2nd revision: See comment 4.

Response 2nd revision: - 

8. Comment 8:

2nd revision: See comment 5. The authors hit the nail, though they could have delivered examples of impacts in this section too.

Response 2nd revision: - 

9. Comment 9:

2nd revision: (Comment 8-9(10)) The authors provided the suggested schematic connection between the nine features and health care. The section is informative now, though it still lacks a synthesis. Maybe its presence – A draw? Flow diagram? - could help readers generalizing the application of the frame. Although the authors commented on the examples more in-depth than in the first submission, maybe the readers could find it challenging to apply the tools to a specific case.

With their thorough revision, the authors improved the first and the second parts of the paper. The final sections get better after the revision, but it requires further work to keep the previous sections' promises.

While reading it, it is hard to focus on the research question, as authors still do not explicitly connect their results’ discussion with keywords like triple aims and system performances, for just quickly quote them in the Limitations and Conclusion sections.

Conclusion and Limitation must exploit the potential that is now shown by the previous sections. At the moment, they seem to be far weaker than the rest of the paper.

Response 2nd revision: We appreciate the reviewer comments. Building on the academic editor comments, we have changed the introduction, methods and analysis sections to clearly state the scientific motivation, research question and objectives so that the results and discussion are in line with what the paper promises. This being said, we have re-structured the discussion and limitations section to better explain the link we have created between SM and IHS performance assessment. We appreciated the idea of a visual aid for better understanding the characterization of SM in the field of IHS, but we think that Table 2, 3 and figure 2 comply with this role. We link the characterization of SM with triple aim and system performance assessment in lines 466-483, when stating the implications of the exposed characteristics of SM.

We think that the best trait of the selection tool is its simplicity. We have changed the writing of the examples to highlight this trait. We hope that the reviewer is pleased with the new discussion and that the contribution of the paper is stated clearly in the conclusion.

10. Comment 10:

2nd revision: See comment 9

Response 2nd revision: -

11. Comment 11:

2nd revision: -

Response 2nd revision: -

Academic Editor 2nd Revision:

1. Comment 1.

As the reviewers suggested, the academic value of the work shall be strengthened. Currently, it is unclear about the research questions and how the state-of-the-art is advanced by this work. The scientific motivation is still missing.

Response: We thank the editor for his revisions. We acknowledge that the introduction, starting with the title, invited readers into a topic that was broader than what the paper is covering. For this reason, we have strengthened the introduction to highlight the scientific motivation and the contribution of our work and clearly stating the real scope of our article. While including a more comprehensive revision of past literature, we have clearly stated the existing knowledge gap (lines 106-110) and the position of the paper.

2. Comment 2.

2. The analysis part shall be strengthened. What are the key messages resulting from the analysis? Also, it would be nice to have this article discussing the research trends and shedding light on future research directions.

Response: We thank the editor for his comment. The new introduction is a better guide into what readers can expect from the paper and goes in line with the analysis. The analysis explanation on the methods section 2.4 was corrected so that the expectations created in the introduction and the results are in line. We agree that the key messages from the analysis should be strengthen, and we do so in the changes we did to the conclusion section. To summarize: 1. SM can use system thinking to evaluate topic of relevance for IHS. 2. SM allow IHS managers to correctly attribute causality, optimize interventions, and create long term assessments. 3. Selecting a simulation technique is facilitated when both the characteristics of the modeling techniques are understood, and the complexities ruling the system performance are identified and prioritized. Our framework for assessing complexity can be used for this purpose.

We hope that with the new introduction section (that highlights the contribution and scientific motivation of our work) it is left clearer why the current analysis is fit for the purpose.

As suggested by the editor, we have included a much broader contextualization, so we can better position the contribution of our paper to current literature.

3. Comment 3.

There have been dozens of literature review / survey papers on simulation models of healthcare applications. The position of this paper is unclear. How is this paper different from the existing review papers?

Response: As suggested by the editor, we have included a much broader contextualization, so we can better position the contribution of our paper to current literature. (lines 76-104)

4. Comment 4.

The number of papers on healthcare simulation is tremendous. Currently, only 27 papers were analyzed. This coverage is much narrow than those covered by the existing review papers. I suggest the authors have a more comprehensive review of the studies, particularly the recent ones. To my knowledge, the below studies are relevant to this review work. However, the list below is not exhaustive, and the authors shall identify further related studies:

1. Abramovich, M. N., Hershey, J. C., Callies, B., Adalja, A. A., Tosh, P. K., & Toner, E. S. (2017). Hospital influenza pandemic stockpiling needs: a computer simulation. American journal of infection control, 45(3), 272-277.

2. Chen, Y., Kuo, Y. H., Balasubramanian, H., & Wen, C. (2015, December). Using simulation to examine appointment overbooking schemes for a medical imaging center. In 2015 Winter Simulation Conference (WSC) (pp. 1307-1318). IEEE.

3. Gul, M., & Guneri, A. F. (2015). A comprehensive review of emergency department simulation applications for normal and disaster conditions. Computers & Industrial Engineering, 83, 327-344.

4. Kuo, Y. H. (2014). Integrating simulation with simulated annealing for scheduling physicians in an understaffed emergency department. HKIE transactions, 21(4), 253-261.

5. Kuo, Y. H., Leung, J. M., Graham, C. A., Tsoi, K. K., & Meng, H. M. (2018). Using simulation to assess the impacts of the adoption of a fast-track system for hospital emergency services. Journal of Advanced Mechanical Design, Systems, and Manufacturing, 12(3), JAMDSM0073-JAMDSM0073.

6. Kuo, Y. H., Rado, O., Lupia, B., Leung, J. M., & Graham, C. A. (2016). Improving the efficiency of a hospital emergency department: a simulation study with indirectly imputed service-time distributions. Flexible Services and Manufacturing Journal, 28(1-2), 120-147.

7. Hu, X., Barnes, S., & Golden, B. (2018). Applying queueing theory to the study of emergency department operations: a survey and a discussion of comparable simulation studies. International transactions in operational research, 25(1), 7-49.

8. Moeke, D., van de Geer, R., Koole, G., & Bekker, R. (2016). On the performance of small-scale living facilities in nursing homes: a simulation approach. Operations research for health care, 11, 20-34.

9. Niessner, H., Rauner, M. S., & Gutjahr, W. J. (2018). A dynamic simulation–Optimization approach for managing mass casualty incidents. Operations research for health care, 17, 82-100.

10. Ordu, M., Demir, E., Tofallis, C., & Gunal, M. M. (2020). A novel healthcare resource allocation decision support tool: A forecasting-simulation-optimization approach. Journal of the operational research society, 1-16.

11. Roy, S. N., Shah, B. J., & Gajjar, H. (2020). Application of Simulation in Healthcare Service Operations: A Review and Research Agenda. ACM Transactions on Modeling and Computer Simulation (TOMACS), 31(1), 1-23.

12. Roy, S., Prasanna Venkatesan, S., & Goh, M. (2020). Healthcare services: A systematic review of patient-centric logistics issues using simulation. Journal of the Operational Research Society, 1-23.

13. Salleh, S., Thokala, P., Brennan, A., Hughes, R., & Booth, A. (2017). Simulation modelling in healthcare: an umbrella review of systematic literature reviews. PharmacoEconomics, 35(9), 937-949.

14. Vanbrabant, L., Braekers, K., Ramaekers, K., & Van Nieuwenhuyse, I. (2019). Simulation of emergency department operations: A comprehensive review of KPIs and operational improvements. Computers & Industrial Engineering, 131, 356-381.

15. Vanbrabant, L., Martin, N., Ramaekers, K., & Braekers, K. (2019). Quality of input data in emergency department simulations: framework and assessment techniques. Simulation Modelling Practice and Theory, 91, 83-101.

16. Weissman, G. E., Crane-Droesch, A., Chivers, C., Luong, T., Hanish, A., Levy, M. Z., ... & Halpern, S. D. (2020). Locally informed simulation to predict hospital capacity needs during the COVID-19 pandemic. Annals of internal medicine, 173(1), 21-28.

17. Yousefi, M., Yousefi, M., & Fogliatto, F. S. (2020). Simulation-based optimization methods applied in hospital emergency departments: A systematic review. Simulation, 96(10), 791-806.

18. Zhang, C., Grandits, T., Härenstam, K. P., Hauge, J. B., & Meijer, S. (2018). A systematic literature review of simulation models for non-technical skill training in healthcare logistics. Advances in Simulation, 3(1), 1-16.

Response: As we mentioned before, starting with the title, the paper was not clear in the scope of the research. We have worked to better explain the scope of our paper, which gives reason to our much smaller sample of papers. We understand that the selection criteria was not detailed correctly, and we have worked to correct this issue (sections 2.1 & 2.2). We gave emphasis on the link between the objectives of the paper and the selection criteria, and why are these appropriate.

In short, we selected papers using a simulation technique in topics of interest to integrated healthcare system that took a system thinking perspective and had the detail and quality necessary to understand the complexities in components relations that were represented. 

We also acknowledge the limitation of our systematic search in terms of the databases that we used. In particular, because articles such as 4.(Kuo 2014) ; 5.(Kuo 2018); 6.(Kuo 2016) & 8. (Moeke 2016) were not present in said databases. We will acknowledge this limitation in the corresponding section of our article(line 538).

In the same line, we acknowledge that our search strategy did not identify all relevant papers on simulation modeling, as we addressed a very specific use case which led to the exclusion of papers such as 1. (Abramovich 2017) and 2. (Chen 2015). Our search strategy with the focus on integrated healthcare was based on the work and expertise of worldwide leaders in the topic of performance assessment of integrated healthcare systems in an effort to focus the characterization of simulation modeling and their ability to implement system thinking for this particular area of research. To attend to this issue, we have strengthened the argument behind our search strategy, and we have stated a clear differentiation to what could be characterized as a comprehensive review of simulation modeling in healthcare management. (lines 126-127 147-151; 161-165; 181)

Given that the, now better explained, aim of our study is a characterization of the field of simulation modeling in areas of interest for integrated healthcare systems (particularly in the discipline’s ability to implement system thinking) and not a quantitative summary in the line of a meta-analysis, we are confident that our methodological approach allowed us to analyze a pertinent set of rigorously selected articles.

We hope that our substantive revision of the manuscript and response to the constructive comments by the reviewers and editor do now fully meet your expectations.

---

## [Decision Letter · Decision Letter 2]

19 Apr 2021

PONE-D-20-10819R2

Simulation modeling to assess performance of integrated healthcare systems: Systematic literature review to characterize the field and visual aid to guide model selection.

PLOS ONE

Dear Dr. Larrain,

Thank you for submitting your manuscript to PLOS ONE. After careful consideration, we feel that it has merit but does not fully meet PLOS ONE’s publication criteria as it currently stands. Therefore, we invite you to submit a revised version of the manuscript that addresses the points raised during the review process.

We look forward to receiving your revised manuscript.

Kind regards,

Yong-Hong Kuo

Academic Editor

PLOS ONE

Additional Editor Comments (if provided):

The revision has been reviewed by the reviewer from the last round. The reviewer has made a favorable recommendation.

I have gone through the revision again. I highly appreciate that the authors have seriously addressed some of the concerns. The scope of the work and research question are now clear.

I believe the work has certain degree of merit and potential to be published. However, since PLOS ONE publishes scientifically rigorous studies, there are still two major issues which have to be addressed:

1. As compared with other similar studies on simulation modeling in healthcare system, the studies reviewed in this work (only 27) are significantly fewer. The arguments and conclusions made in this study are therefore not promising. This problem is particularly clear as shown in Section 3.2. For examples, the results presented are only based on 2 studies for Markov models, 1 study for microsimulation, 2 studies for agent-based models, 1 study for hybrid simulations, etc. What were claimed in those sections are not comprehensive and convincing.

2. The problem stated in point 1 is probably caused by a subjective selection of articles (on p. 10) by only the two authors. This work is title a "systematic" review but this process systems to be non-systematic. I suggest there is a clear description of how the reviewers determine whether to include a study is included. A quality score is given to an article by the two authors; however, is this quality score reliable? (Do the authors imply that out of 2271 articles, only 27 are of quality? Others are not of quality?) Another issue is that a systematic literature review is to identify trends in the research area. It would be necessary to include studies not only based on the subjectively determined quality but need to identify the overall trends.

Reviewers' comments:

Reviewer's Responses to Questions

**Comments to the Author**

1. If the authors have adequately addressed your comments raised in a previous round of review and you feel that this manuscript is now acceptable for publication, you may indicate that here to bypass the “Comments to the Author” section, enter your conflict of interest statement in the “Confidential to Editor” section, and submit your "Accept" recommendation.

Reviewer #1: All comments have been addressed

2. Is the manuscript technically sound, and do the data support the conclusions?

Reviewer #1: Yes

3. Has the statistical analysis been performed appropriately and rigorously? 

Reviewer #1: N/A

4. Have the authors made all data underlying the findings in their manuscript fully available?

Reviewer #1: Yes

5. Is the manuscript presented in an intelligible fashion and written in standard English?

Reviewer #1: Yes

6. Review Comments to the Author

Reviewer #1: Is the manuscript technically sound, and do the data support the conclusions?

The authors have progressively addressed the original version issues. Their paper now sounds informative while it illustrates its aims and methods sharply. The authors illustrate their framework clearly. Now the flow is logical, and connections between scope, data, and conclusions are evident.

Have the authors made all data underlying the findings in their manuscript fully available?

Yes, they did.

Is the manuscript presented in an intelligible fashion and written in standard English?

The authors have progressively evolved their manuscript through discourse with the reviewers. After the last revision, the paper will offer a novel and informative perspective to health care scholars and especially professionals. It will help readers look at the “Triple Aim” model with a broader framework to assess health care performances considering complexity accurately and robustly. Though the proposed tool has limits, it shows the readers the relevance of the systemic approach in evaluating a health care system. The paper starts a practical conversation that goes beyond simple improvement in health care assessment techniques. This proposal falls in the field of accurate and relevant feedback conversation (e.g, see Zollo, 2009 about superstitious learning). Despite its limits, the tool connects the diverse elements that concur in performance and allows H.C. professionals to take them and their complex interactions into account in evaluating and comparing performance.

This reviewer can't judge the English level of the manuscprit.

7. PLOS authors have the option to publish the peer review history of their article (what does this mean?). If published, this will include your full peer review and any attached files.

Reviewer #1: **Yes: **Andrea Montefusco

---

## [Author Response · Author response to Decision Letter 2]

1 Jun 2021

Dear reviewer and Academic Editor:

We are deeply thankful for your revision. We have revised each of your comments and provide each with a response in the following text. For a more structured version of this response, please see the attached “Response to Reviewers" file.

Editor Comments, General:

The revision has been reviewed by the reviewer from the last round. The reviewer has made a favorable recommendation.

I have gone through the revision again. I highly appreciate that the authors have seriously addressed some of the concerns. The scope of the work and research question are now clear.

I believe the work has certain degree of merit and potential to be published. However, since PLOS ONE publishes scientifically rigorous studies, there are still two major issues which have to be addressed:

- Response:

We are deeply thankful to the Editor for recognizing our work in the article. We have made extensive changes to the core of the article to address the editors’ concerns. Regarding the Editor’s second comment, in summary, we understand that the current selection of articles is deemed to be more subjective than what is usually found in systematic literature reviews. In particular, because even though the criterion for selection is clearly defined, part of the definition (Section 2.2 line 155: <<Finally, we excluded from the data extraction and analysis studies whose reporting standards were insufficient to fully understand and replicate the assessment >>) leaves room for a subjective interpretation. While even in the most rigorous systematic reviews data extractors have to introduce a certain level of interpretation in deciding which study and which data to include, we agree that our approach previously has not been optimal. In order to correct for subjectivity, we have now included all the articles with the highest quality score following our quality assessment (+3 extra articles). Furthermore, to be transparent with reader expectations, we have decided to define our article as a literature review with systematic a search . The PROSPERO record was modified stating this change. We have noted that PLOS ONE publishes all types of reviews of the literature (in terms of Grant and Booth typology), not only systematic reviews of the literature. However, by now labelling our study as literature review with systematic search we believe that our methodological approach meets all expectations a reader might have.

Editor Comments, 1.

1. As compared with other similar studies on simulation modeling in healthcare system, the studies reviewed in this work (only 27) are significantly fewer. The arguments and conclusions made in this study are therefore not promising. This problem is particularly clear as shown in Section 3.2. For examples, the results presented are only based on 2 studies for Markov models, 1 study for microsimulation, 2 studies for agent-based models, 1 study for hybrid simulations, etc. What were claimed in those sections are not comprehensive and convincing.

-Response:

We thank the editor for the comment. Even though we defend our article selection to be fit for purpose and aligned with the objective of the study, we recognize that the writing of section 3.2 leads readers to think that the revision of the different methodologies was based solely in the articles selected from the systematic search. This is not correct. The methodologies were extracted from the selected articles. Then, selected articles were used to explain each methodology features. Nevertheless, the features themselves were summarized using complementary literature (presented in table 2). This is explained in the methods section 2.4 (line 184-186). We have changed the first paragraph of the section 3.2 (lines 223-228) to better explain were the features explanation come from. The complementary literature used is not solely based on case articles, but mostly on review articles and books explaining the methodologies in detail.

Editor Comments, 2.

2. The problem stated in point 1 is probably caused by a subjective selection of articles (on p. 10) by only the two authors. This work is title a "systematic" review but this process systems to be non-systematic. I suggest there is a clear description of how the reviewers determine whether to include a study is included. A quality score is given to an article by the two authors; however, is this quality score reliable? (Do the authors imply that out of 2271 articles, only 27 are of quality? Others are not of quality?) Another issue is that a systematic literature review is to identify trends in the research area. It would be necessary to include studies not only based on the subjectively determined quality but need to identify the overall trends.

-Response:

First, please see above regarding the change in the title of the document according to Grant´s and Booth’s typology.

Second, we edited the methods section, stating explicitly that articles were selected according to our appreciation of their reporting standards to allow for the objective of our paper(section 2.2). In other words, the final selection criteria relate to the ability of reviewers to fully understand and replicate the article for a complete assessment of how the complexities of the system were included in the model. The limitations section was also edited stating the limitation of the selection process.

Three extra articles were added following the correction in the selection process:

1. Ansah et al. 2016: The article was initially excluded because even though evaluated with quality A, it did not add any additional information to the analysis. Following the corrected exclusion criteria, the article is now added in the review.

2. Goldman et al. 2004: The article was not included in the original revision because even though the model has top quality, the practical application and validation of said model are found in subsequent articles. Meanwhile, articles depicting practical applications of the model lacked a detailed description of the model. Initially, we though evaluating them together as a whole was a reach of our quality assessment criteria, hence the initial exclusion.

3. Comans et al. 2017: Similarly, to number (1.), the paper was excluded because it did not add information for analysis. Following the corrected exclusion criteria, the article is now added in the review.

We included the assessment of the new articles and mention them as examples to explain system thinking where they contributed to the analysis. Nevertheless, the results, discussion and conclusion did not change (with exception of the count of optimization capabilities and time frame for assessment) with the inclusion with the new articles.

The abstract was edited showing these changes.

We understand that many systematic reviews indicate research trends, but we are certain that research trends are only reported when in line with the objectives of the study. Our article did not focus on pooled effect sizes or research trends, as the explanation of how SM methods are able to integrate system thinking is not time dependent. In the same line, the selection of a SM method is determined by identifying complexities and the overall objective and not on trends.

Furthermore, given that we have chosen to classify the article as a literature review (where reporting research trends is less common), we think that reporting research trends is less justified.

Reviewers' comments:

All comments by the reviewer were positive and no detailed response is needed.

-Response:

We thank the reviewer for the recognition of our work, and we appreciate his positive appreciation and useful comments during the revision process.

---

## [Editor Report · Decision Letter 3]

28 Jun 2021

Simulation modeling to assess performance of integrated healthcare systems: Literature review to characterize the field and visual aid to guide model selection.

PONE-D-20-10819R3

Dear Dr. Larrain,

We’re pleased to inform you that your manuscript has been judged scientifically suitable for publication and will be formally accepted for publication once it meets all outstanding technical requirements.

Kind regards,

Yong-Hong Kuo

Academic Editor

PLOS ONE
---

## [Editor Report · Acceptance letter]

1 Jul 2021

PONE-D-20-10819R3 

Simulation modeling to assess performance of integrated healthcare systems: Literature review to characterize the field and visual aid to guide model selection. 

Dear Dr. Larrain:

I'm pleased to inform you that your manuscript has been deemed suitable for publication in PLOS ONE. Congratulations! Your manuscript is now with our production department. 

Kind regards, 

on behalf of

Dr. Yong-Hong Kuo 

Academic Editor

PLOS ONE